



# The movement of atmospheric blocking systems: can we still assume quasi-stationarity?

Jonna van Mourik[1,2,3], Hylke de Vries[2], and Michiel Baatsen[3]

[1]Department of Physical Geography, Faculty of Geosciences, Utrecht University, Utrecht, The Netherlands
[2]Royal Netherlands Meteorological Institute (KNMI), De Bilt, The Netherlands
[3]Institute for Marine and Atmospheric Research, Utrecht University, Utrecht, The Netherlands

**Correspondence:** J. van Mourik (j.vanmourik1@uu.nl)

**Abstract.** The quasi-stationary behaviour of atmospheric blocking is studied using a Lagrangian framework that enables the tracking of blocks in space and time. By combining a blocking index based on geopotential height with a lagrangian tracking algorithm, we investigate the characteristics of atmospheric blocking events for different zonal propagation velocities and their impacts on surface temperatures within the retuned EC-Earth3 global climate model. We observe that blocking events can portray a large variety of propagation velocities. Distinct differences are found between the behaviour of eastward-moving blocks and westward-moving blocks, both in size and in spatial distribution. Although the size of blocks is of bigger importance for the temperature anomalies, the propagation velocity has an influence on the strength of the temperature anomalies in winter, due to the slower mechanism of air advection in winter, compared to diabatic heating in summer. In summer, the propagation velocity primarily influences the positioning of temperature anomalies relative to the centre of the blocking system. These findings highlight the complex interactions between size, propagation velocity, and other blocking attributes, and their influence on temperature anomalies. Further research is warranted to explore regional differences in blocking behaviour and impact, as well as how atmospheric blocking and associated temperature anomalies may evolve under future climate conditions.

## 1 Introduction

Atmospheric blocking is a large-scale atmospheric process, where a persistent Rossby wave causes a strong high-pressure area to remain in place (Rex, 1950a, b; Platzman, 1968; Altenhoff et al., 2008). The influence of blocking events on our weather is significant as these blocks cover a large area, can last from days up to a month, and are quasi-stationary (Liu, 1994). Atmospheric blocking in winter is often associated with cold spells, which often form downstream of the block. Upstream of the block, a warm conveyor forms with warm air and moist conditions. Both these upstream and downstream processes are caused by horizontal advection of respectively warm air from the tropics and cold air from the polar regions, driven by the high-pressure area. In summer, heatwaves and droughts are often associated with persistent conditions of atmospheric blocking. During the day, diabatic warming and adiabatic warming due to subsidence reinforce each other and result in positive temperature anomalies (Bieli et al., 2015; Röthlisberger and Martius, 2019). As subsidence plays a bigger role in the realisation of higher temperatures in summer, these temperatures are found right underneath the block (Kautz et al., 2022).



The dynamics of atmospheric blocking has been a subject of research for over a century, predominantly due to the high
impact weather associated with them (Garriott, 1904). Since then, scientists have been trying to understand different blocking
characteristics that attribute to the impact of blocks on our weather, such as their size (Nabizadeh et al., 2019), intensity
(Wiedenmann et al., 2002; Davini et al., 2012), duration (Barnes et al., 2012), and location (Brunner et al., 2018; Sousa
et al., 2017). Other topics that have been widely researched, are the dynamics behind blocking formation and maintenance.
Orography and land-sea contrasts have long been named as important contributors to blocking formation (Ji and Tibaldi, 1983).
More recently, the release of latent heat during cloud formation has been evaluated by Pfahl et al. (2015), while Yamazaki and
Itoh (2013) proposed the selective absorption mechanism as a contributor to the persistence of blocking.

An often overlooked aspect of atmospheric blocking is the propagation velocity of the block. Because of its quasi-stationary
nature, the velocity component is often ignored for simplicity and it is not included in blocking indices that are used to
identify blocks from datasets (Sousa et al., 2021; Pelly and Hoskins, 2003). Sumner (1963) was the first to introduce the
subject of blocking propagation velocity, making the distinction between progressive (eastward moving), quasi-stationary, and
retrogressive (westward moving) blocks. These categories were split by a threshold value of $5°$ per day in both the eastward
and westward direction, and it was found that about $40\%$ of all blocks was quasi-stationary, $35\%$ was progressive, and $25\%$ was
retrogressive for the Atlantic-European section. However, as Sumner (1963) used a minimum blocking duration of two days,
these values are probably not representative for our current understanding of blocks with a minimum duration of four days.
More recently, Mokhov and Timazhev (2019) showed that different threshold values of maximum zonal propagation velocities
led to different frequencies of blocking. Lastly, Steinfeld et al. (2018) studied the distributions of the mean zonal propagation
velocity during the lifetime of blocks and concluded that this velocity was highest in their onset phase and decreased during
their mature phase. They also mention that their blocking algorithm detected some blocks with propagation velocities larger
than $10\text{ m s}^{-1}$ ($864\text{ km d}^{-1}$), but they doubted if they could be seen as classical blocks.

To our knowledge, no studies have considered the effect that the propagation velocity of atmospheric blockings has on our
weather. This may, in part, be attributed to the prevalence of the use of an Eulerian framework in the examination of atmospheric
blocking, which lacks the capacity for tracking and analyzing the movement of atmospheric blocks. However, considering that
the development of winter temperature extremes is a process that unfolds gradually (Brunner et al., 2017), it is plausible
that blocking propagation velocities influence the impact caused by the block. Properly assessing the potential impact of
their movement is thus a key element of understanding the impacts from atmospheric blockings. In our research, we make a
first attempt to deepen this understanding using the latest update of the climate model EC-Earth3$_{p5}$ (ECE3p5), focusing on
the Northern Hemisphere, and using a Lagrangian perspective. Working from a Lagrangian perspective enables us to track
the blocks in space and time, and gather information on individual blocks. This perspective offers a more dynamic view of
blocks, enabling us to discern variations in propagation velocities and their corresponding influences on weather phenomena—a
capability that remains elusive within an Eulerian framework. Our study has two objectives: to start with, we will assess the
characteristics of the zonal propagation velocity of atmospheric blocks and how it relates to blocking size, intensity, duration,



and location; after that, we will look at the influence of the propagation velocity on temperature anomalies and compare this to the influence of blocking size in order to estimate the importance of the propagation velocity of atmospheric blocking.

## 2 Data and Methods

### 2.1 EC-Earth3p5 model

In this study, we use simulations of the EC-Earth3$_{p5}$ climate model (ECE3p5), which is the latest update of EC-Earth3 (ECE3). One of the main shortcomings of ECE3 is its temperature bias compared to ERA5 reanalysis data. In the Northern Hemisphere, ECE3 has a cold bias, while in the Southern Hemisphere a warm bias dominates. Compared to ECE3, ECE3p5 has an even larger warm bias over the Southern Hemisphere, but with a better representation of the temperature over the Northern Hemisphere (Muntjewerf et al., 2023; Van Dorland et al., 2023)(Appendix 5). As we are only evaluating the Northern Hemisphere atmospheric blocking, ECE3p5 is the most appropriate choice for this research. Later on in the method, the ability of ECE3p5 to simulate atmospheric blocking is evaluated.

From ECE3p5 we use the historical dataset, which contains 16 ensemble members ranging from 1850 to 2014, resulting in a total dataset of 2624 years. These 16 ensemble members were created by perturbing the initial state of the system, thus giving rise to 16 climate realisations. The standard resolution is T255L91 ($\approx 80$ km) for the atmosphere, and the model uses time steps of 2700 s (45 minutes). For the sake of computation time and because of the large size and duration of atmospheric blocks, we regrid this dataset to a regular lat-lon grid of $2.5° \times 2.5°$ and take the daily mean values. For the identification of the blocks, the geopotential height at 500 hPa (Z500) (m) is used. The other selected variable is the daily mean surface air temperature at 2m (TAS) (K).

### 2.2 ERA5 reanalysis data

ERA5 reanalysis data is used to evaluate the ECE3p5 model. From this dataset, we only use the daily mean geopotential at 500 hPa ($m^2 s^{-2}$), which is transformed to the geopotential height (hPa)(Copernicus Climate Change Service, 2023). ERA5 has a resolution of $0.25° \times 0.25°$ (Hersbach et al., 2023), but in order to comply with our regridded ECE3p5 dataset, we regrid the $0.25° \times 0.25°$ grid of ERA5 to $2.5° \times 2.5°$.

### 2.3 Blocking index

Since Rex (1950a) first introduced a definition of atmospheric blocking, multiple blocking indices have emerged to extract these blocks from datasets. They differ in the dynamical aspects of a block that they grasp, but also in the minimal duration, size, or location of a block. Some indices only capture one dimensional aspects, while others caption multiple dimensions. As a result, it is difficult to compare different studies on blockings (Sousa et al., 2021; Davini and d'Andrea, 2020; Barnes et al., 2014).





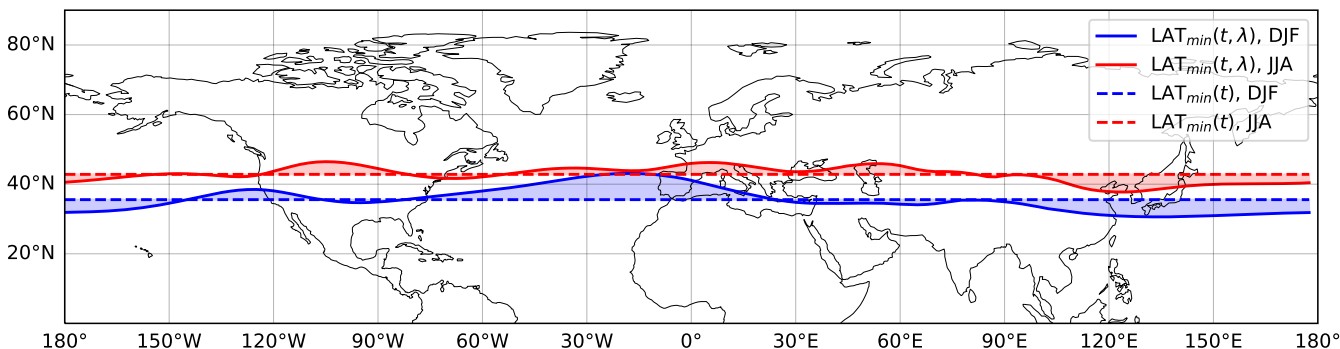

**Figure 1.** Difference between the latitude-dependent minimum latitude $\text{LAT}_{min}(t,\lambda)$ (continuous line), and the spatial-averaged minimum latitude $\text{LAT}_{min}(t)$ (dashed line). Both shown for winter (DJF, blue), and summer (JJA, red).

The blocking index that we use here, is based on the two-dimensional extension of the standard geopotential height-index of Tibaldi and Molteni (1990) and takes the majority of the alterations of Sousa et al. (2021) into account, which are meant to make the index more inclusive and precise. The first step of Sousa et al. (2021) is to exclude the subtropical high pressure belt from the blocking analysis. Where Sousa et al. (2021) uses a daily mean value for the minimum latitude for all longitudes $\text{LAT}_{min}(t)$, we chose to work with a longitude dependent minimum latitude $\text{LAT}_{min}(t,\lambda)$, which looks for the latitude at which the value of Z500 is higher than the Z500 value averaged over the previous 15 days:

$$Z500(\lambda,\phi,t) > \overline{Z500}. \tag{1}$$

As the location of the subtropical high depends on the position of the sun and thus on the seasons, the minimum latitude is found further poleward in summer compared to winter. Especially over the wintertime Euro-Atlantic region, the choice of $\text{LAT}_{min}$ makes a difference, since the longitude dependent $\text{LAT}_{min}(t,\lambda)$ is situated $7.5°$ further poleward than $\text{LAT}_{min}(t)$, which is shown in Figure 1. Above the minimum latitude, the blocks are filtered out using these geopotential height gradients

$$\text{GHGS}(\lambda,\phi,d) = \frac{[Z500(\lambda,\phi,d) - Z500(\lambda,\phi-\Delta\phi,d)]}{\Delta\phi} \text{ if } \text{LAT}_{min} \leq \phi, \tag{2}$$

$$\text{GHGN}(\lambda,\phi,d) = \frac{[Z500(\lambda,\phi+\Delta\phi,d) - Z500(\lambda,\phi,d)]}{\Delta\phi} \text{ if } \text{LAT}_{min} \leq \phi \leq 75°, \tag{3}$$

$$\text{GHG}(\lambda,\phi,d) = \sqrt{\text{GHG}_z{}^2 + \text{GHG}_m{}^2}, \tag{4}$$

where we define the zonal gradient and the meridional gradient respectively as

$$\text{GHG}_z = \frac{Z500(\lambda,\phi+\Delta\phi,d) - Z500(\lambda,\phi-\Delta\phi,d))}{2\Delta\phi}, \tag{5}$$



$$\text{GHG}_m = \frac{Z500(\lambda + \Delta\lambda, \phi, d) - Z500(\lambda - \Delta\lambda, \phi, d))}{2\Delta\lambda}, \tag{6}$$

with $\Delta\lambda = \Delta\phi = 2.5°$. Following Sousa et al. (2021), a grid cell is said to be in a blocked state when GHGN<0, GHGS>0, and GHG$(\lambda, \phi, t) < 20$ m per degree. Furthermore, blocks are excluded whenever their size is smaller than $5 \cdot 10^5$ km$^2$ and their duration is shorter than four days, with no differentiation made among various block types as is done by Sousa et al. (2021). Lastly, we apply the method of Wiedenmann et al. (2002), which was modified by Davini et al. (2012) to be two-dimensional, in order to derive a quantification of blocking intensity (BI):

$$\text{RC}(\lambda, \phi, d) \equiv \frac{(Z_u + Z500(\lambda, \phi, d))/2 + (Z_d + Z500(\lambda, \phi, d))/2}{2\phi}$$

$$BI(\lambda, \phi, d) = 100[\frac{Z500(\lambda, \phi, d)}{RC} - 1.0]. \tag{7}$$

In Equation 7, RC stands for representative contour, $Z_u$ is the minimum value within $60°$ upstream of a value Z500 and $Z_d$ is the minimum value within $60°$ downstream of this same Z500. The value resulting from it, indicates how the meridional circulation is affected by the presence of the block, where northern hemisphere blocks are defined as weak when they have a value of BI<2.0, moderate when 2.0<BI<4.3, and strong when BI>4.3. Wiedenmann et al. (2002) based this categorisation on which blocks were within and outside of one standard deviation of the 30-year mean intensity of their dataset. Applying the described method leads to the climatological blocking intensities shown in Figure 2, which can be compared to the results from the CMIP6 models in Davini and d'Andrea (2020). The blocking climatologies of ERA5 and ECE3p5 differ in their mean blocking intensity in certain regions, which are more prominent during the separate seasons than for the annual mean. An over- or underestimation can be caused both by an over- or underestimation of the amount of blocks and by its intensity. Annually, ECE3p5 underestimates the BI over western Europe and Scandinavia, and overestimates the BI over Alaska and Siberia. In winter, BI is generally higher than the annual mean, with two dipoles to be found in the comparison between the model and reanalysis data: one with underestimation over western Europe and overestimation over the Ural region, and one with an underestimation over the Bering Sea and an overestimation over Alaska. In summer, lower BI are found, together with the same dipoles for the differences between ERA5 and ECE3p5, but switched in sign. These dipoles are further discussed in Section 4.1.

### 2.4 2d Cell-tracking Algorithm

As we want to evaluate the propagation velocity of atmospheric blocks, we work from a Lagrangian point of view. Using the 2D cell-tracking algorithm of Lochbihler et al. (2017), we can find continuous cells and track them in time. Continuous cells are formed from adjacent grid cells that meet our blocking index. Continuous cells belong to one track when there is overlap between the cells during consecutive time steps. The algorithm results in information on the size, duration, intensity (both mean and maximum), weighted center, and dates of onset and decay. From this information, blocks are selected with a minimum





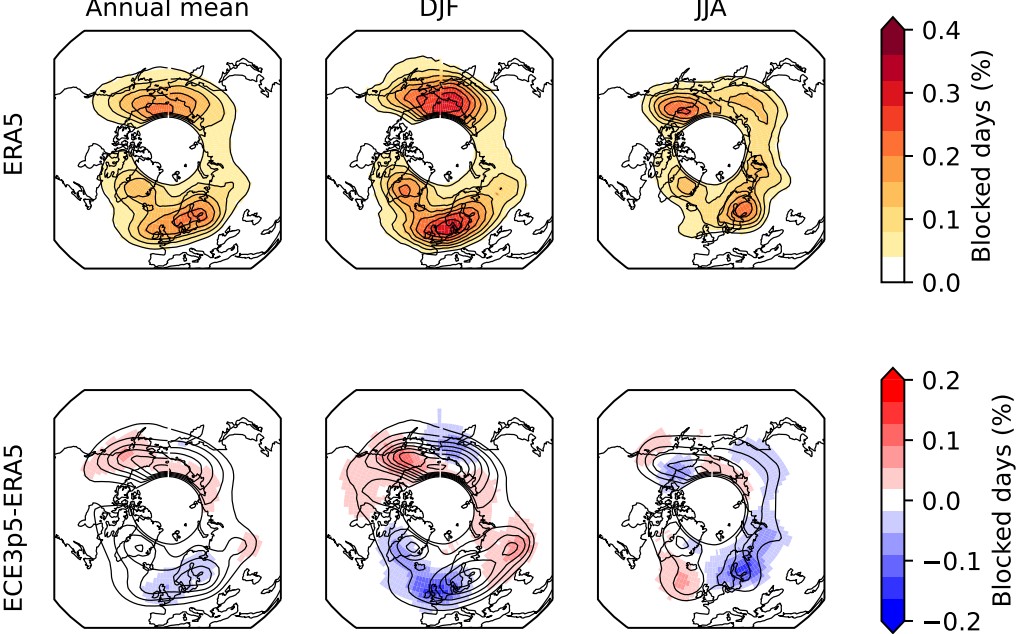

**Figure 2.** Upper row: Climatological blocking intensity ERA5 (contours and shading) for the annual mean, winter mean (DJF), and summer mean (JJA). Lower row: contours show blocking intensity of ECE3p5, shading shows the difference between ERA5 and ECE3p5 for the annual mean, winter mean, and summer mean. Both the data from ERA5 and ECE3p5 are taken over the period of 1951-2014.

135      duration of four days. The size of the blocks is converted from a number of grid cells to km$^2$, using the latitude of the intensity weighted center as $\phi$ in

$$dx \approx 2\pi R \cos\phi \frac{2.5°}{360°}, \tag{8}$$

where $R$ is the radius of the Earth and $2.5°$ the size of our grid cells. After small and short-lived blocks are removed, the propagation velocity of the remaining blocks is calculated. As the primary moving direction of the blocks is zonally, we only

140      take the zonal propagation velocity into account. This velocity is calculated by comparing the intensity-weighted center of a block at its date of onset to its date of decay and divide this distance in kilometers by the blocking duration. This method is consistent with the propagation velocity used by Steinfeld et al. (2018). Blocks are categorised as winter or summer blocks based on their onset date being in respectively the months December, January or February, or June, July or August. A summary of this complete method is showcased in Figure 3.





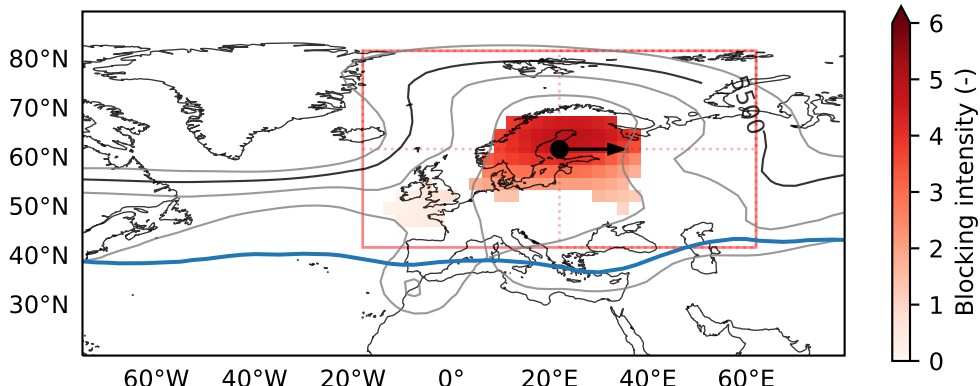

**Figure 3.** Atmospheric block filtered out by the method of Sousa et al. (2021), but with a variable minimum latitude (blue line), and added blocking intensity (red) (Wiedenmann et al. (2002)), tracked by a celltracking algorithm. The black dot denotes the weighted centre of the block, and the arrow the zonal velocity and direction. The contour lines show the geopotential height (z500) for 5400, 5500 (black), 5600 and 5700 m. The red square is the area used to evaluate the temperature, divided in 4 subareas.

## 2.5 Impact on temperature

To assess the impact of different zonal propagation velocities on our weather, we look at the daily mean surface air temperature at 2m, detrended and corrected for climatology to get temperature anomalies. These temperature anomalies allow us to compare the impact of blocking on different locations and during different seasons. As blocks do not only influence the temperature right underneath the block, but also in the areas around it, we work with a fixed area around the blocking centre. This area is set to have a latitude-width of $40°$ and a longitude-width of $80°$ (see red square in Figure 3). The size of this area is selected to encompass the size of the majority of the blocks, ensuring that it adequately captures the diverse range of temperature anomalies associated with them. This size is comparable with approximately a quarter of the area shown in Figure 3. We divide the $40° \times 80°$ area around the block into four quadrants to incorporate the different temperature anomalies upstream and downstream of a block. We only take into account those temperature anomalies that are measured over land, as the ocean has a significantly larger thermal warming capacity compared to land (Cess and Goldenberg, 1981). Consequently, it takes longer for the surface area underneath the block to warm or cool over the ocean. At the same time, the temperature effects of blocks are primarily experienced by us on land. Therefore, the land surface temperature anomalies are of particular relevance when considering the impacts of blocking events.



# 3 Results

## 3.1 Comparison between ERA5 and ECE3p5

We evaluate the accuracy of ECE3p5 to simulate blocking behaviour by comparing the output of the tracking algorithm for ECE3p5 to the output for ERA5. This is done for the amount of blocks per year, their size, duration, average intensity, maximum intensity, and absolute velocity. This information is summarised in Table 1 for the annual and seasonal (winter/summer) values of blocks within ECE3p5 and ERA5 after aggregating over the entire Northern Hemisphere.

Variables that differ significantly between ERA5 and ECE3p5 according to the $t$-test are shown in bold. Comparing all variables shows that the only significant differences can be found in the annual mean and summer mean amount of blocks, where ECE3p5 simulates less blocks than can be observed in ERA5. From these values, it is likely that the difference in the annual mean, which has 10 more cases in ERA5, is caused by the difference in the summer mean, which has 7 more cases in ERA5. However, we can't exclude any contributions from spring and autumn, as these seasons are not taken into account. The difference between the amount of summer and winter blocks is also bigger for ERA5 than for ECE3p5. All other variables do not show significant differences, possibly due to the relatively large standard deviations indicating the big differences between individual blocks. Both for ERA5 and ECE3p5, winter blocks are generally larger, more intense, and reach higher absolute zonal velocities than summer blocks, while their durations are about the same. For the size and duration it should be noted that our minimum requirement was respectively $5 \cdot 10^5$ km$^2$ and four days, as described in Section 2.3. Not shown in Table 1 is the direction of the velocities, which can be both eastward and westward. We found the annual mean direction to be positive, meaning that on average atmospheric blocks are moving towards the east, despite the general conception that blocks are quasi-stationary. This same eastward trend was also observed in a case study by Steinfeld et al. (2020) using potential vorticity (PV) anomalies, but not yet on this scale for all observed blocks in reanalysis and model data.

## 3.2 Relation between propagation velocity and other characteristics

Table 1 only results in seasonal averages per blocking characteristic. We extend this analysis to all winter and summer blocks separately, which allows us to study any correlations between the different blocking characteristics and the propagation velocity. In Figure 4, the propagation velocity is plotted against the size, the duration, and the average intensity of the blocks, both for summer and winter. Next to the kernel density estimation (KDE, in shading), the $10^{\text{th}}$, $50^{\text{th}}$, and $90^{\text{th}}$ percentiles are used to mimic the behaviour of the two extreme states and the mean state of the characteristics per velocity bin of 40 km/day. Moving towards more extreme velocities, the results get noisier due to the few number of cases.

The size is shown in the first column of Figure 4. From the KDE, it is noticeable that the blocking size is much more confined to smaller sizes in summer, while the spread is larger in winter. The majority of the blocks in both winter and summer are on the smaller side compared to the total spread and are associated with eastward velocities. On the contrary, the largest blocking sizes are associated with westward velocities. This uneven allocation becomes even clearer when looking at the percentiles,





| | # Blocks (yr$^{-1}$) | Size ($\cdot 10^6$ km$^2$) | Duration (d) | BI$_{av}$ (-) | BI$_{max}$ (-) | Velocity (km d$^{-1}$) |
|---|---|---|---|---|---|---|
| **ERA5** | | | | | | |
| Annual mean | **117(9)** | 1.1(1.0) | 5.8(2.3) | 1.8(1.0) | 2.4(1.2) | 295(225) |
| Winter | 24(4) | 1.5(1.3) | 5.8(2.2) | 2.2(1.2) | 3.0(1.5) | 315(251) |
| Summer | **36(4)** | 1.0(0.8) | 6.0(2.5) | 1.4(0.7) | 1.8(0.8) | 268(205) |
| **ECE3p5** | | | | | | |
| Annual mean | **107(8)** | 1.3(1.1) | 5.8(2.3) | 1.9(1.0) | 2.5(1.3) | 310(246) |
| Winter | 25(4) | 1.6(1.3) | 5.7(2.1) | 2.2(1.2) | 3.1(1.4) | 354(279) |
| Summer | **28(4)** | 1.1(0.9) | 6.0(2.5) | 1.4(0.7) | 1.9(0.9) | 264(205) |

**Table 1.** Blocking statistics over the period of 1951-2014 for ERA5 (above) and ECE3p5 (below). Significant differences between ERA5 and ECE3p5 are shown in bold. Numbers in the table denote the mean and (standard deviation). The standard deviation is taken over all blocks within the specified season over all 16 ensemble members. The velocity is the absolute zonal velocity of the blocks.

which represent the 10% smallest, the 10% largest, and the median blocking size per velocity bin. It is consistently observed across both seasons that the larger the block, the faster it moves westward. Towards the quasi-stationary blocks, the blocking size decreases. In winter, the blocking size stays approximately the same for different eastward propagation velocities, while in summer the blocking size slightly increases again. This result is similar to the behaviour of Rossby waves according to the Rossby wave theory, where the group velocity is westward for larger waves and eastward for smaller waves (Holton and Hakim, 2013). The similarity in behaviour between blocking and Rossby waves stresses the link between the two phenomena and might be of use in predicting certain characteristics of a block.

The duration of the blocks is plotted against the propagation velocity in the second column of Figure 4. The KDE and the percentiles for the duration are not as smooth compared to those for size or intensity. This disparity arises from the discrete nature of duration measurements, quantified in whole days, in contrast to the continuous values of size and intensity. The KDE is pyramidal for both winter and summer, with the longest durations over the stationary blocks. In both seasons, blocks with shorter durations occur more frequently than blocks with longer durations. The 90[th] percentile shows for both winter and summer that stationary blocks have a longer duration than the faster propagating blocks. The 50[th] percentile, which shows the mean value per bin, is five days for almost all velocities and therefore does not show the same relationship as the 90[th] percentile line. The 10[th] percentile is a straight line at four days, which is the minimum duration threshold that we set in Section 2.3.

The final blocking characteristic that is compared to the propagation velocity is the average blocking intensity in the third column. The first notable difference between winter and summer is the spread in the data, as shown by the KDE. The winter months exhibit a much larger variability and higher values than the summer months. Both seasons display an oval distribution around slightly eastward moving velocities, with a denser core for lower intensities. This indicates that the more stationary blocks display the largest range in blocking intensities. For the percentiles in winter, only the 90[th] percentile shows this same




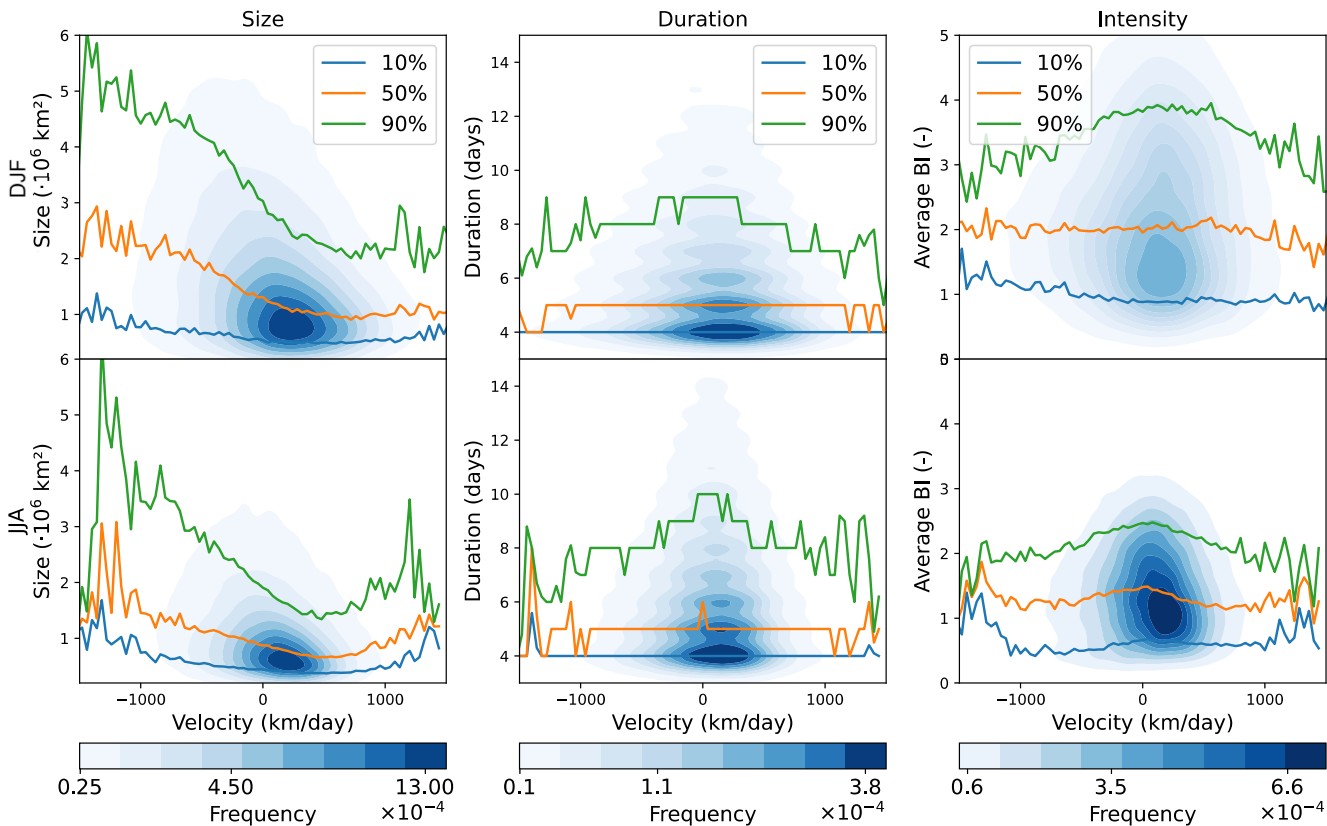

**Figure 4.** Kernel density estimation (blue shading, linear) and the 10th (blue), 50th (orange), and 90th (green) percentiles for the size (first column), duration (second column), and intensity (third column), all for winter (DJF, upper row) and summer (JJA, lower row). Taken over the period of 1850-2014 over all 16 ensembles of ECE3p5.

210  relationship of larger intensities for quasi-stationary blocks. The 50th and 10th percentiles do not show this, and stay around the same BI no matter the propagation velocity, which can also be deduced from the broader base of the KDE. In summer, the 90th percentile show a larger BI for quasi-stationary blocks and lower strengths for faster moving blocks in both directions, although combined with a lot of noise at the outer boundaries. The sign in the 50th and 10th percentiles is minimal, just as for winter, which suggests that blocking intensity is likely not dependent on the propagation velocity.

215  Combining all the information in Figure 4 tells us that quasi-stationary blocks are generally smaller, have a longer duration, and have a broad variation in strength; eastward-moving blocks are generally larger than quasi-stationary blocks, have a



shorter duration, and a smaller variation in strengths; lastly, westward-moving blocks are bigger than both quasi-stationary and eastward-moving blocks, are of shorter duration, and have a smaller range of strengths than quasi-stationary blocks.

**Figure 5.** Blocking characteristics for (left) westward and (right) eastward moving blocks. The characteristics shown are the mean size (blue), the duration (yellow), the average intensity (green), and the absolute propagation velocity (red) over the years of 1850-2014 and all 16 ensembles. The thicker lines are the 15 day rolling means, and the thinner lines show the mean values over all blocks and ensembles per day of the year. The shading is the standard deviation of the 16 ensemble members.

### 3.3 Seasonality of westward- and eastward-moving blocks

220  Westward-moving blocks and eastward-moving blocks thus seem to be different, especially when it comes to their size. To extend our analysis on the different aspects of blocking behaviour further, we assess the seasonality of westward- and eastward-moving blocks separately. This is done in Figure 5 for the size, duration, average intensity and velocity per day of the year,



with the westward-moving blocks on the left and the eastward-moving blocks on the right. Note that the velocity is positive for both eastward and westward blocks and thus represents the amplitude and not the direction of the velocity.

225 The blocking propagation velocity (red in Figure 5) is the first variable to be examined and later compared to the other variables. Both the eastward and the westward velocities have a seasonal cycle with higher values in winter, a decrease in the propagation velocity in spring, a minimum in summer, and an increase in velocity again in autumn. Notably, there are discernible differences between the two. On average, eastward-moving blocks tend to have higher propagation velocities than their westward counterparts across all seasons. However, these disparities diminish during the transition from spring to summer and towards 230 the end of winter. Both have their maximum velocities in winter, with a value of $373 \pm 39$ km d$^{-1}$ for the eastward blocks and $364 \pm 42$ km d$^{-1}$ for the westward blocks, where the variability is given as one standard deviation over all ensemble members and years on that specific day of the year. The eastward-moving blocks have their maximum at the end of December, while the westward-moving blocks have their maximum at the beginning of February. However, as the eastward-moving blocks have roughly the same velocity throughout the whole winter, this maximum is more spread out than for the westward-moving blocks. 235 Their minimum velocities also differ, as the eastward velocities have their lowest values in June ($261 \pm 39$ km d$^{-1}$), while the westward velocities have their lowest values in July and September ($230 \pm 42$ km d$^{-1}$). Over the year, both blocking types thus exhibit roughly the same seasonality, but with shifts in their minima and maxima of one to two months, combined with larger values for eastward-moving blocks.

After the division of all atmospheric blocks over westward-moving and eastward-moving blocks, the other blocking charac- 240 teristics are also split up according to their zonal direction, leading to different sizes (blue), durations (yellow), and intensities (green) for the westward-moving blocks in Figure 5a and the eastward-moving blocks in Figure 5b. When we split up the size of the blocks with respect to their propagation direction, a clear distinction arises between the size of the blocks that move to the west and that move to the east. Over the seasons, westward-moving blocks are consistently larger than eastward-moving blocks, with the disparities being more pronounced in winter compared to summer. Both have their maximum size at the end 245 of winter, with values of $1.4 \pm 0.2 \cdot 10^6$ km$^2$ for the eastward variant and $2.1 \pm 0.4 \cdot 10^6$ km$^2$ for the westward variant. Their minimum size is found at the end of summer, with values of $0.8 \pm 0.2 \cdot 10^6$ km$^2$ for the east and $1.0 \pm 0.4 \cdot 10^6$ km$^2$ for the west. Apart from the differences in size, there are no clear differences in the seasonality between the two blocking directions. This leads to an interesting misalignment or lag of two months between the minimum velocity and the minimum size for eastward propagating blocks, while the minimum size of the westward propagating blocks can be found more in the middle of its 250 velocity minima.

The next variable is the duration of the blocks. As indicated in Table 1, the variations across seasons are minimal. A similar observation holds true for the two blocking directions, where only marginal differences exist among the two. Both the eastward and westward blocks have the same values for their minima and maxima, with a minimum value of $5.6 \pm 0.2$ days and a maximum value of $6.1 \pm 0.2$ days. This results in a difference of $0.5 \pm 0.2$ days between the minima and the maxima of the



blocks. Within these small differences, both blocking directions have their maximum duration in July and their minima in winter.

The final variable is the average blocking intensity, which reveals a distinct seasonality that is approximately consistent for blocks moving in both directions. Both have a constant average intensity over the winter and parts of spring and autumn, which then rapidly declines in summer. The only difference seems to be that the transitions between the steady state in winter and

the minimum value in summer is more abrupt for the westward-moving blocks compared to the eastward-moving blocks. Both have minimum values of $1.3 \pm 0.3$ in July and maximum values of respectively $2.3 \pm 0.3$ and $2.2 \pm 0.3$ in winter for eastward and westward blocks. Over the year, the average intensity is higher during seasons where blocks also have a higher propagation velocity, and lower in seasons with lower propagation velocities.

### 3.4  Spatial distribution of different blocking velocities

Categorising blocks into velocity based groups leads to further insights on the different behaviour of westward- and eastward-moving blocks. When ordering the zonal velocities from negative to positive, a division is made between the first $10\%$ ($v_x \leq P10$), which are the fastest westward-moving blocks, and the last $10\%$ ($v_x \geq P90$), which are the fastest eastward-moving blocks, and all blocks in between ($P10 < v_x < P90$). For each group, the coordinates of the weighted centres of the blocks based on their BI on the fourth day of their existence is used to show where the different groups of blocks occur most. The

choice of the fourth day aligns with the division made by Steinfeld et al. (2018) between the onset phase (day 1 and 2) and mature phase (day 3, 4 and 5) of the block. By focusing on the fourth day, we ensure that we assess the blocks during their mature phase. This analysis was performed for both winter and summer, resulting in Figure 6. It is important to note that the $10\%$ fastest eastward- and westward-moving blocks do not have the same velocities in winter and summer, as is evident from the distinct distributions in Figure 4. Additionally, it should also be kept in mind that the distribution shown in Figure 6 is only

a representation of the centers of the blocks and not their entire spatial extent.

The spatial distribution differs per group of blocks. The majority of the blocks are represented by the $P10 < v_x < P90$ group (middle column). In winter, these blocks are most commonly found over Greenland, Alaska, and northeast Siberia, and in lower amounts stretching from western Europe into Russia. Comparing this spatial pattern to the $10\%$ fastest westward moving blocks ($v_x \leq P10$, first column), we can see that the hotspot over Europe is now less present and shifted north. Greenland is

also less featured for the P10 blocks, while especially the hotspot over northeast Siberia has a way higher frequency compared to the average blocking pattern. The $10\%$ fastest eastward moving blocks ($v_x \geq P90$, third column) show the exact opposite tendency. The hotspots over Alaska and northeast Siberia are a little less featured for the P90 blocks, while the pattern over Europe has intensified and also shifted southeastward. The hotspot over Greenland shows the same pattern en frequency for the P90 blocks as the average spatial pattern. Overall, we can say that northeast Siberia is the preferred location for faster westward

moving blocks, while faster eastward moving blocks are more commonly found over middle Eurasia.





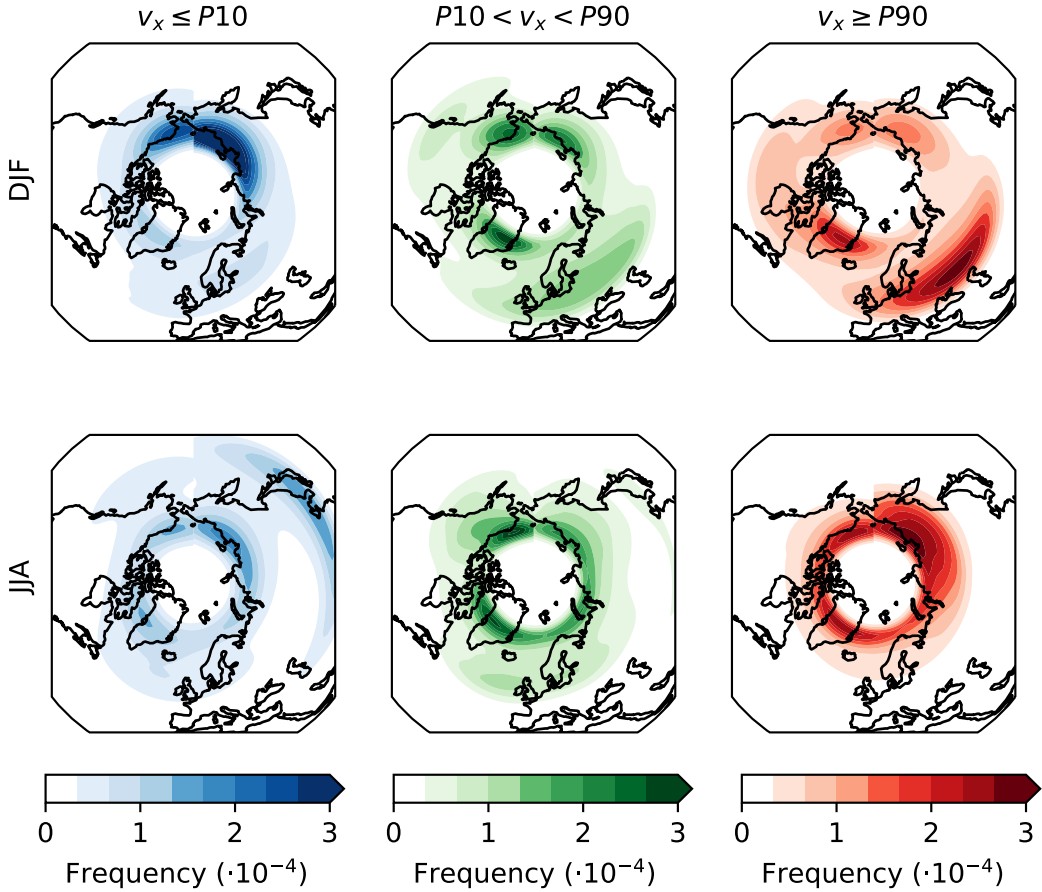

**Figure 6.** Spatial distribution weighted (based on BI) blocking centres, on the fourth day. First column: the $10\%$ fastest westward-moving blocks ($v_x \leq P10$). Middle column: all propagation velocities in between ($P10 < v_x < P90$). Third column: the $10\%$ fastest eastward-moving blocks ($v_x \geq P90$) All taken separately over all winter months (DJF, upper row) and over all summer months (JJA, lower row) over the period of 1850-2014 for all 16 ensembles of ECE3p5.

The spatial distribution of summer blocks is quite different from the winter distribution. As the middle column for the $P10 < v_x < P90$ blocks shows, summer blocks are usually more confined to the higher latitudes. Higher frequencies are found over Greenland and Alaska, although blocks are quite evenly distributed between those areas. P10 blocks occur more often over Siberia than over Alaska and Greenland, similar to the winter circumstances. The spatial pattern of the P10 blocks deviates from the mean at lower latitudes in Asia, which seems to be just above our minimum latitude. These could consist of anomalously large ridges moving towards the west, which might still by captured by our blocking index. The P90 blocks on the other hand, resemble the mean blocking pattern quite closely. There is a strong confinement to the higher latitudes, with an even distribution between the North Sea and the Bering Sea. The highest frequencies are found over Siberia, which is also the





region with the largest deviations compared to the mean situation. Although the differences are smaller in summer compared
to winter, it is clear from Figure 6 that blocks that have different propagation velocities also reside in different areas. The
underlying mechanisms for this observation remain to be studied.

## 3.5 Influence of the propagation velocity on temperature

Within the examining of the relationship between various blocking characteristics and the propagation velocity, Figure 4
revealed that quasi-stationary blocks tend to have longer durations and higher intensities, making them more likely to have
a significant impact on our weather. On the contrary, the fastest westward-moving blocks exhibited larger surface areas, while
quasi-stationary blocks were generally smaller in size. Considering that both larger and more stationary blocks are expected to
contribute to higher temperature anomalies, it is important to determine the respective contributions for these factors. First, we
compare temperature anomalies with different propagation velocities, and subsequently we assess the impact of blocking size
compared to velocity on temperature anomalies.

To investigate whether the propagation velocity has an effect on the temperature anomalies resulting from the block, we look
at the results for the lower-right quadrant (see Section 2.5) in Figure 7. The other quadrants can be found in the Supplements,
Figures S8, S9, and S10. As the quadrants mostly differ in the strength of the anomalies but not so much in the patterns related
to the propagation velocities, we choose to discuss just one of them here.

The temperature anomalies over land are plotted against the propagation velocities in Figure 7 for winter and summer. In the
KDE plot in winter, a broad range of temperature anomalies is observed, ranging from $-10°$C to $10°$C. The majority of blocks
result in relatively small temperature anomalies, as is indicated by both the $50^{th}$ percentile, which is around zero degrees
for most velocities. This is in line with our earlier results in Figure 4, where it was shown that the majority of the blocks
are relatively small, of short duration, and relatively weak. These blocks result in lower temperature anomalies, moderating
the mean values. Simultaneously, the coldest temperature anomalies are also measured for stationary blocks, indicating a
correlation between lower temperatures and stationary blocking conditions. This result is in line with the research done by
Brunner et al. (2017), who showed that cold spells take longer to develop due to the slow process of horizontal advection of cold
air. Conversely, the warmest temperature anomalies can be associated with the faster westward-moving blocks, which creates
an interesting contrast between the blocking velocities associated with the coldest temperatures. The pattern of the warmest
$10\%$ temperature anomalies looks similar to the relation between size and propagation velocity in Figure 4, so possibly the
larger block size associated with faster westward-moving blocks plays a role in these larger temperature anomalies. Although
winter blocks are mostly related to cold spells, this figure clearly demonstrates that these winter blocks can generate a wide
range of temperature anomalies, not limited to negative values.

In contrast to the winter conditions, the temperature anomaly differences in summer exhibit a more constrained distribution.
The KDE plot displays smaller excesses, the $50^{th}$ percentile remains about zero, and the $10\%$ warmest and coldest temperature
anomalies are just a little higher than $\pm2.5°$C. Notably, none of the percentiles exhibit a discernible trend, as their values





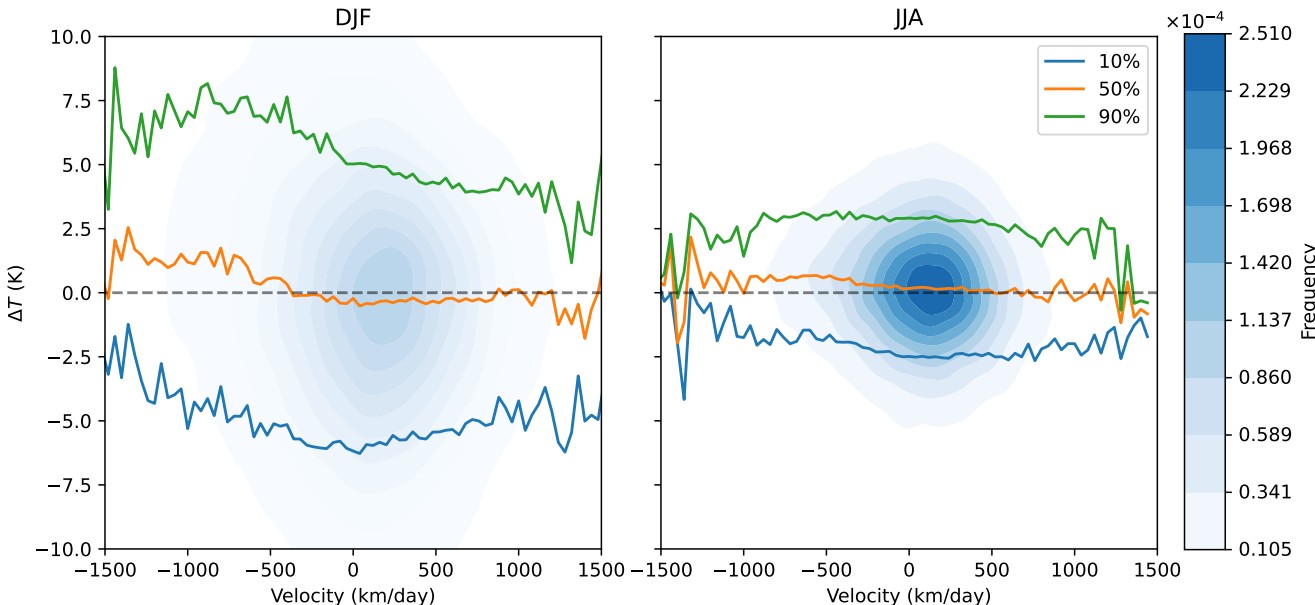

**Figure 7.** Kernel density estimation (blue shading) and the $10^{th}$ (blue), $50^{th}$ (orange) and $90^{th}$ (green) percentiles of the lower-right quadrant of the 2m temperature anomaly plotted against the propagation velocity, all for winter (DJF, left) and summer (JJA, right). Only temperature anomalies over land are taken into account. Taken over the period of 1950-2014 over all 16 ensemble members of ECE3p5.

remain relatively consistent across the entire range of propagation velocities, with just a little smaller anomalies towards the extreme velocities, although clouded by the noise in the signal. The insensitivity of the temperature anomalies with respect to the propagation velocities can be attributed to the different warming processes in summer, caused by both diabatic and adiabatic warming, both faster processes than air advection such that the warming effect can still take place beneath faster-

moving blocks. Especially for Rex blocks, this warming underneath the block goes hand in hand with cooling underneath the accompanying lower pressure area to the south of the block. These lower temperatures are represented by the 10% lowest temperature anomalies in Figure 7. The relatively small temperature anomalies observed in summer compared to winter can be attributed to multiple factors. Firstly, the data used for this analysis consists of daily mean temperatures. The cloud-free conditions that cause warming underneath these summer blocks during the day, also cause cooling during the night. These

lower night temperatures moderate the overall daily mean values. To quantify this effect, it would be necessary to study the differences in response between the minimum and maximum temperatures. Secondly, as argued by Cheung et al. (2013), the continent is already warmer in summer, leading to lower temperature anomalies. Additionally, pressure gradients are smaller in summer compared to winter. This last argument can be compared to the blocking intensity that we studied in Figure 5, where indeed summer blocks generally exhibit lower blocking intensities compared to winter blocks.

In order to assess the different effects of propagation velocity and size on temperature anomalies, a composite analysis is performed in which all blocks are divided according to their size and propagation velocity. This result is shown in Figure 8



for temperature anomalies measured over land. On the x-axis, the propagation velocities are sorted from the largest negative values (0%) to the largest positive values (100%). The y-axis shows the size per 20%, from small (0%) to large (100%). For both winter and summer, the quasi-stationary blocks with a velocity closest to zero are found at roughly 40%. The choice to use percentiles and not absolute differences in size or propagation velocity was made to ensure that every composite mean has enough cases for a reliable result. As a consequence, the percentiles do not represent the same velocity or size values for winter and summer, as the percentages are taken separately for the two seasons. The values that the percentages represent are shown in Table 2. Figure 8 shows the composite mean analysis for winter (a) and summer (b) over land. Focusing on winter, it is

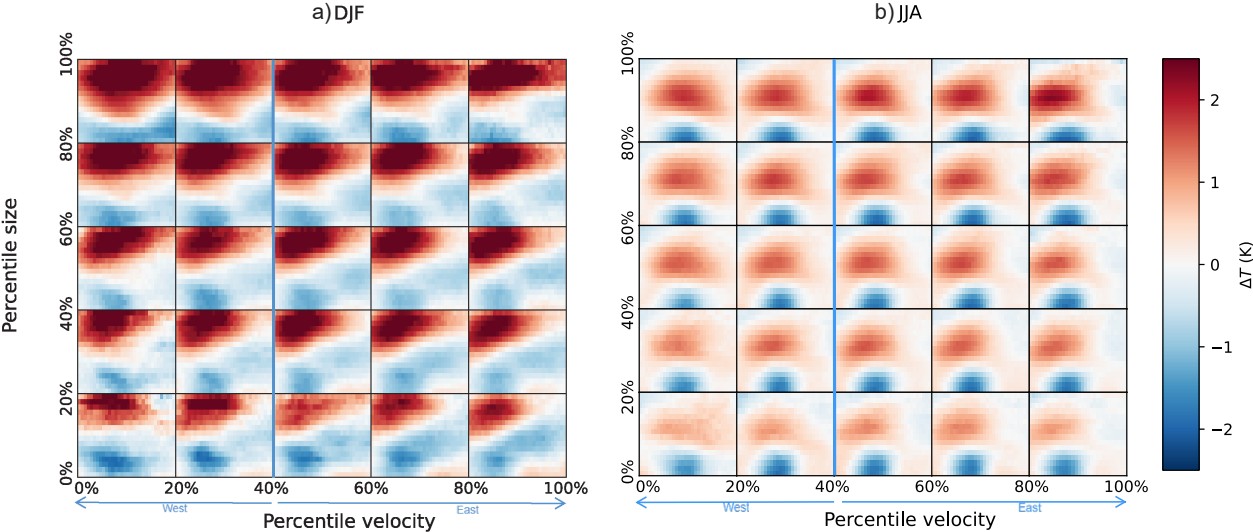

**Figure 8.** Composite mean of 2m temperature anomalies in a region of $40° × 80°$ around the blocking centre over land. The winter season (DJF) is shown in (a), and the summer season (JJA) in (b). On the x-axis, the size is divided by its percentiles, going from small to large. On the y-axis, the propagation velocity is divided by its percentiles, going from negative values to positive values. The blue line indicates the division between westward-moving blocks and eastward-moving blocks, also indicated by the arrows. Data is taken from ECE3p5 over the period of 1850-2014 and over all 16 ensemble members.

|  | 0% | 20% | 40% | 60% | 80% | 100% |
|---|---|---|---|---|---|---|
| **Size$(\cdot 10^6 \mathrm{km}^2)$** |  |  |  |  |  |  |
| Winter | 0.20 | 0.94 | 1.26 | 1.59 | 2.05 | 8.35 |
| Summer | 0.20 | 0.65 | 0.83 | 1.03 | 1.32 | 5.51 |
| **Velocity$(\mathrm{km/day})$** |  |  |  |  |  |  |
| Winter | −1718 | −121 | 42 | 168 | 317 | 1709 |
| Summer | −1709 | −75 | 38 | 133 | 249 | 2031 |

**Table 2.** Blocking sizes and propagation velocities for different percentiles in winter and summer.





evident that for every combination of size and propagation velocity, a distinct pattern emerges. Positive temperature anomalies
are consistently observed to the northwest of the block, while negative temperature anomalies are observed to the southeast of
the block. For all of them, the positive temperature anomalies dominate in value over over the negative temperature anomalies.
When we compare blocks with different sizes for the same propagation velocity, the temperature anomalies become stronger
and take over a larger part of the selected area around the block. This result is to be expected, as a larger block will influence
the temperature over a larger area. Comparing blocks with different propagation velocities but with the same size reveals a
change in the temperature pattern. In the case of westward-moving blocks, the surface area surrounding the block exhibits a
larger proportion of positive temperature anomalies, primarily concentrated to the north of the block, extending towards the
blocking centre. The faster eastward a block moves, the more confined the positive temperature anomalies are to the northwest,
leaving more room for the negative temperature anomalies.

For summer in Figure 8b, the dipole structure typical for Rex blocks is clearly visible. Each combination of size and propagation
velocity has a warm core around the centre of the block, and a cold core to its south. Just as in winter, the temperature anomalies
get stronger for larger blocks, although this seems to primarily affect the warm core. For the same size but different velocities,
the temperature anomaly pattern barely changes, which is what we expected based on the faster warming mechanisms in
summer. The only notable change is a zonal shift of the dipole with respect to the blocking centre. For westward-moving
blocks the dipole is located at the blocking centre, while for the fastest eastward-moving blocks the dipole is located more on
the west-side of the block. Not only the location of the dipole as a whole changes, but it also tilts more towards the east for
westward-moving blocks and towards the west for westward-moving blocks. For both directions, the warm core is thus located
on the upstream side of the block as a result of the warming that has already taken place there.

## 4 Discussion on methodology

### 4.1 Model biases

The results in Section 3 need to be considered in light of the model accuracy and choices that were made for the blocking
index. In Section 2.1 it was described that the ECE3p5 model was chosen as it simulates more realistic temperatures for the
Northern Hemisphere compared to ECE3. In Figure S1b, the warming trend in both ERA5 and ECE3p5 can be seen for the
Northern Hemisphere, where the warming trend in ECE3p5 is stronger than for ERA5. However, this trend does not seem
to significantly change the model outputs for the propagation velocity of atmospheric blocks over the years, as can be seen
in the Supplements, Figure S5. Therefore, the differences in temperature trends are probably of minimal importance for the
propagation velocity, although they may be important for other blocking characteristics (Woollings et al., 2018; Nabizadeh
et al., 2019). At the same time, a positive temperature anomaly caused by a block will affect us more in a warmer climate,
which makes for a new research topic on impacts of blocks under global warming.

Temperature is also closely linked to the geopotential height. Higher temperatures lead to an expansion of the lower atmosphere
and thus to a higher geopotential height (Christidis and Stott, 2015). This relation between temperature and geopotential height





is evident in both the model and the reanalysis data, as depicted in Figure S2, but with an overestimation of the geopotential height of ECE3p5. In addition to this overestimation of the geopotential height, the model also tends to overestimate its gradients, as indicated in Figure S3, where the regions of underestimation are identical to the regions where the model underestimates the blocking frequency in Figure 2. The correlation between the overestimation of Z500 and the blocking frequency

is less prominent, but can still be recognized. These findings suggest that the model's biases in Z500 and its gradients may contribute to its limitations in accurately representing blocking events.

Other biases that play a role in most climate models, are overestimation of the jets over the oceans (Anstey et al., 2013; Delcambre et al., 2013; Pithan et al., 2016), and misrepresentation of the orography (Narinesingh et al., 2020; Berckmans et al., 2013). Although we did not test if those biases are also incorporated in the ECE3p5 model, it is plausible to think so.

Models often simulate jets to be too strong and too far land-inwards. As blocks form at the end of those jets, this results in an eastward shift of the blocking frequency in models. This same effect can be seen for the winter simulations of ECE3p5 in Figure 2, where the model underestimates the amount of blocks over the Atlantic and the Pacific, while there is an overestimation over Russia and Alaska. The orography plays an important role in mountainous areas, such as the Ural and Alaska. These regions both show a negative bias in the amount of simulated blocks by ECE3p5.

## 4.2 Choice of blocking index

As discussed in Section 2.3, there are many different blocking indices used by researchers. These differences in the definition and calculation of blocking indices can have implications for the outcomes of our study. Here, we examine some of these discrepancies and how they may affect our results.

One significant modification we made compared to the method proposed by Sousa et al. (2021) was the redefinition of the

minimum blocking latitude, as depicted in Figure 1. This adjustment of the minimum blocking latitude resulted in a mean poleward shift of the minimum latitude over the Atlantic and a mean equatorward shift over the Pacific ocean. These shifts have implications for the identification and measurement of blocking events. When we use a fixed minimum latitude, we get the blocking frequency as shown in Figure S6 in the Supplements, where indeed more ridges are captured than in Figure 2. Consequently, this divergence also impacts the outcomes presented in Figure 6. Using the more static minimum latitude

of Sousa et al. (2021) leads to Figure S7. In this figure, the blocks appear to be concentrated along the minimum latitude, particularly during the summer season. This shows the extent in which our results are influenced by the chosen minimum blocking latitude.

Some studies employ a second geopotential height gradient to further refine the selection of blocking events. This additional gradient,

$$\mathrm{GHGS}_2(\lambda, \phi, d) = \frac{[Z500(\lambda, \phi - \Delta\phi, d) - Z500(\lambda, \phi - 2\Delta\phi, d)]}{\Delta\phi}, \tag{9}$$

is also used by Sousa et al. (2021), but they use it solely to differentiate between Rex blocks and Omega blocks. Keeping the additional gradient would thus lead to a loss of all blocks that do not have the strong flow reversal underneath the block, such





as omega blocks and ridges. This additional gradient is also used by Davini and d'Andrea (2020). The results that Davini and d'Andrea (2020) get for their blocking frequencies resembles our result in Figure 2, even though they used a fixed minimum latitude of 30°N. Their use of GHGS$_2$, which restricts their blocks mostly to Rex blocks that are found at higher latitudes, presumably allows for the use of a lower and fixed minimum latitude.

A last important aspect of our study is the utilization of blocking centres, which are tracked using the 2D celltracking algorithm. As outlined in Section 2.4, the blocking centre is determined by weighting the blocking intensity. Considering our minimum blocking area of $500000 \ km^2$, the blocking centre has the potential to move a considerable distance while remaining within the boundaries of the blocked area. Despite this, we opted to work with the weighted centre rather than an unweighted centre due to the association of higher intensities with greater impacts. As some of the blocks had long tails with a very low intensity, the unweighted centre would have trailed too far away from the area that is impacted by the blocks.

## 4.3 Other simplifications

Our study involves more simplifications that may impact the analysis and results. One notable simplification is the treatment of all blocks as equal events, despite their spatial variations, as demonstrated in Figure 6. Blocks occurring at different locations can lead to varying impacts due to factors such as orography, orientation to large water bodies, and latitude, all of which can influence the temperature and humidity of the advected air. While we attempt to compensate for this simplification by separating blocks over land and over oceans, a lot of details will be lost due to the generalization.

Another simplification we made is the focus on winter and summer months. This decision is based on the distinct characteristics of blocks during these seasons, and they are also the most studied in literature. However, it is important to recognize that even within these seasons, there are variations that are averaged out when analyzing on a seasonal basis, as illustrated in Figure 5. The transitional character of spring and autumn makes these seasons more of a challenge to assess. As Brunner et al. (2017) found, the average relation between blocks and their temperature impacts changes during spring, which makes it even more interesting for research in a warming world.

Lastly, the choice to work with a Lagrangian framework for studying the temperature influences of blocking events is not immediately intuitive, especially not when you are interested in the weather directly overhead. However, this approach does offer valuable new insights into how blocking events lead to different temperature anomalies, and which factors contribute to them. It also reveals that the impact of a block is not confined solely to the region directly underneath stationary blocks, as faster moving blocks leave a trace of temperature anomalies as well. Generally, we can thus expect a faster moving block to impact any region that it will cross in the near future. To obtain a more comprehensive picture, it would be beneficial to project the Lagrangian temperature profiles to specific areas of interest. This would enable a better understanding of how blocks, whether they are slowly travelling or stationary, influence temperature patterns in specific regions. By integrating the local-scale projections, we would be able to use both the advantages of a Lagrangian framework and the practical implications for local weather conditions.





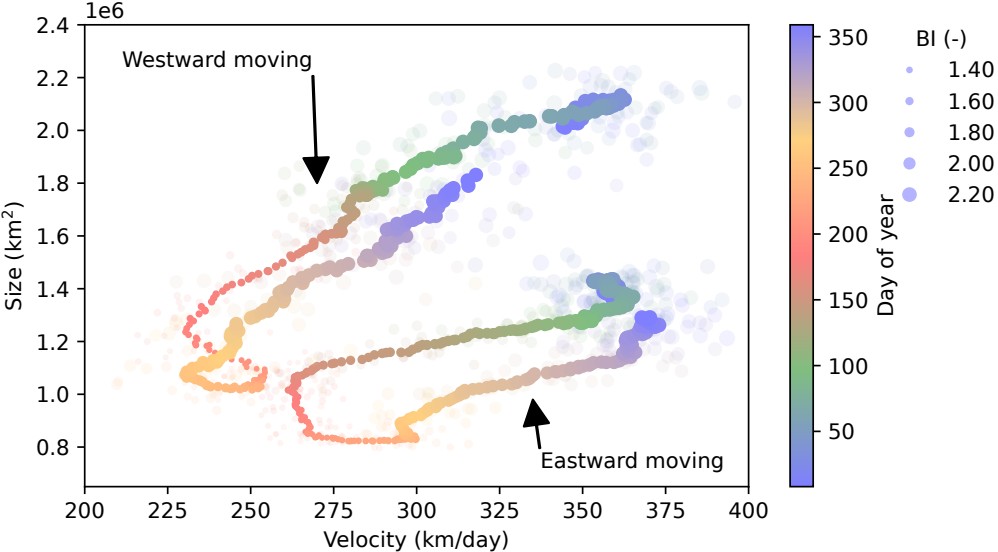

**Figure 9.** Climatological values per day of the year for size (km$^2$) against absolute values of the propagation velocity (km/day). The scatter size indicates the blocking intensity (-), and the colour the day of the year. Each day of the year is averaged over the period of 1850-2014 over all 16 ensembles (light colours), and a 15-day running mean is plotted (opaque colours).

## 5 Conclusions

In conclusion, this study aims to advance our understanding of the propagation velocity of atmospheric blocking and its influence on temperature anomalies at the Earth's surface. Atmospheric blocking has been extensively studied since the 1950's (Rex, 1950a), with a lot of focus on its frequency and impact on our weather in different seasons (Kautz et al., 2022). However, the propagation velocity of atmospheric blocking has been largely overlooked, primarily due to the general quasi-stationary nature of blocks and the predominant use of Eulerian framework in their analysis. Previous studies that did consider the propagation velocity treated it as a secondary aspect, without delving into its significance (Sumner, 1963; Steinfeld et al., 2018; Mokhov and Timazhev, 2019). Steinfeld et al. (2018) concluded its study on blocking propagation by mentioning that some blocks portrayed larger propagation velocities of 10 m s$^{-1}$ (864 km/day), but that perhaps they could not be seen as classical blocks.

To address this research gap, this study adopts a Lagrangian perspective to gain deeper insights into the dynamics of atmospheric blocks. By combining the blocking index of Sousa et al. (2021), the blocking intensity of Wiedenmann et al. (2002), and the celltracking algorithm of Lochbihler et al. (2017), together with some alternations where needed, we employ a unique methodology to identify and track the blocks. With this combination of methods, we evaluate the performance of the EC-Earth3$_{p5}$ climate model compared to ERA5 reanalysis data, examine the seasonality of the zonal propagation velocity and its



interplay with other blocking attributes, and investigate the influence of the propagation velocity on temperature anomalies in comparison to blocking size.

The evaluation of the model performance reveals differences in the spatial distribution of blocking frequency and intensity between ERA5 and ECE3p5. Especially in summer, ECE3p5 underestimates the amounts of blocks over large regions in the Northern Hemisphere, such as over Scandinavia, Alaska, and parts of Russia. This underestimation over Europe is a well-

known limitation of climate models (Woollings et al., 2018). However, when considering frequency, size, duration, intensity and propagation velocity collectively, the only significant difference between ERA5 and ECE3p5 was observed in the summer frequency.

After the evaluation of the ECE3p5 model, we examine the characteristics of atmospheric blocks, focused on their seasonality, and explore relationships between different blocking aspects. Our findings reveal a wide range of propagation velocities,

spanning from -1500 km/day to +1500 km/day, as determined by our blocking index. These values surpass those reported by Steinfeld et al. (2018), who questioned the classification of blocks exhibiting similar velocities. Additionally, we identify seasonality in the size, the duration, the blocking intensity, and the propagation velocity, with duration showing the least pronounced seasonal variation and the others exhibiting clearer patterns. The size, intensity, and propagation velocity exhibit their minimum values in summer, albeit with some phase-shifts among them, while the duration reached its maximum during

this season.

Another noteworthy discovery is the disparity in behaviour between eastward-moving blocks and westward-moving blocks, which is summarised in Figure 9. On a seasonal basis, westward-moving blocks display larger sizes and a more linear relationship between size and velocity compared to the eastward-moving blocks. The observed relationship resembles Rossby wave theory, wherein the largest blocks are associated with westward propagation velocities, while smaller blocks tend to exhibit

eastward propagation velocities. In contrast, the duration and intensity have their maximum for quasi-stationary blocks. Furthermore, we find that the fastest westward-moving and eastward-moving blocks occupied distinct locations compared to the majority of the blocks, and that these locations also varied across seasons. In winter, the fastest eastward-moving blocks are predominantly found over Greenland and stretching from Eastern Europe into Russia, while in summer, they are located over Siberia. The fastest westward-moving blocks are situated over Siberia in both winter and summer.

Investigating the influence of the propagation velocity on surface temperature anomalies, temperature anomalies are found to be larger in winter than in summer. This could partly be attributed to the use of daily mean temperatures, which are tempered in summer due to lower values at cloud-free nights, and to the already warmer continent in summer, leading to smaller anomalies. In winter, the coldest temperatures are associated with quasi-stationary blocks, while the warmest temperatures are associated with westward-moving blocks. On the contrary, no clear relations are found during the summer. This can be explained by the

different warming and cooling mechanisms in winter and summer, which are slower in winter and thus require more time to create larger temperature anomalies, while their effect is almost immediate in summer. We find that even the faster propagation

velocities contribute to temperature anomalies, indicating that there is no need strictly adhere to the assumption of quasi-stationary blocking systems when regarding temperature anomalies, and thus no velocity limit has to be implemented in the blocking index.

Upon categorising the blocks according to their size and propagation velocity, we observe distinct temperature patterns during winter and summer. In winter, the composite mean of all blocks exhibits warmer temperature anomalies to the northwest of the block and colder temperature anomalies to the southeast. In summer, the composite mean reveals warmer temperature anomalies beneath the block and colder temperature anomalies to its south. Comparing these patterns to the mean state of the blocks, we find that larger blocks induced more intense temperature anomalies, while smaller blocks resulted in less pronounced

temperature anomalies. The propagation velocity plays a role in determining the temperature anomaly patterns. In winter, faster eastward-moving blocks lead to more confined warm anomalies, and faster westward-moving blocks to a larger spread of warm anomalies. In summer, the size have the same effect as in winter, but the propagation velocity only influence the location of the warm and cold anomalies relative to the block. Temperature anomalies for faster eastward-moving blocks are situated to the left of the block, while for faster westward-moving blocks, they appear to the right. In both cases, the largest temperature

anomalies are observed upstream of the block, where the block already had the opportunity to warm the surface area below it.

For future work, it would be interesting to investigate this subject further on a more localised scale to discern regional differences more effectively. Questions on why certain type of blocks are more confined to specific regions than others remains to be investigated. Additionally, exploring how the propagation velocity may change under global warming and its potential implications for temperature anomalies would be of great interest. Understanding these dynamics could enhance our understanding

of the evolving impacts of atmospheric blocking in a changing climate.

*Code and data availability.* Will follow

*Competing interests.* The authors declare that they have no conflict of interests.

*Acknowledgements.* Will follow



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
