# Peer review of "On the movement of atmospheric blocking systems and the associated temperature responses"

_EGUsphere, 2024_

## Referee Comment (RC1)

Review result of "The movement of atmospheric blocking systems: can we still assume quasi-stationarity? Mourik et al."

Overall recommendation: Major revision or rejection.

The topic of this manuscript is interesting. However, there are many ambiguous issues in this manuscript. In this manuscript, the authors emphasized the difference of the impact between eastward- and westward-moving ones. In fact, these differences are obvious and natural. In this study, the authors repeated previous studies and results. Thus, no enough new results are found in this manuscript. The authors should further discuss the relationship among the size, movement speed and strength of blocking because they are not independent each other in a blocking system. The authors also ignored many previous similar studies in the introduction and text in this manuscript so that the authors said "**To our knowledge, no studies have considered the effect that the propagation velocity of atmospheric blockings has on our weather**" in the introduction. Such a description is completely misleading. Moreover, I do not think that the author's 2D Cell-tracking Algorithm on the zonal propagation velocity of atmospheric blocking is correct. Thus, I recommend a major revision or even rejection.

Major comments:

(1) Misunderstanding of the quasi-stationarity of blocking.

In fact, the quasi-stationarity of atmospheric blocking as planetary-scale waves is said relative to the movement of synoptic-scale weather systems. The quasi-stationarity does not mean that blocking is not moving. Atmospheric blocking is often classified into three types: stationary or quasi-stationary, westward- and eastward-moving.

(2) **Lines 45-46**: The description on "**To our knowledge, no studies have considered the effect that the propagation velocity of atmospheric blockings has on our weather**" is not correct. The impact of the propagation velocity of atmospheric blockings on local weathers or short-term variability of Arctic sea-ice has been widely investigated in previous studies. For example, Chen and Luo (2017, GRL) and Yao et al. (2017, JC) examined different impacts of westward-moving and

stationary (sometimes, referred to as quasi-stationary) blocking events over Greenland and Ural region on continental cold anomalies or weathers. Then, Chen et al. (2018, JC) further classified Ural blocking into quasi-stationary, westward- and eastward-moving Ural blocking and examined the impact of the three types of Ural blocking on the short-time variability of Arctic sea-ice and continental cold anomalies. Zhang and Luo (2020) also examined how the Arctic sea-ice decline over the west of Greenland influences the zonal propagation velocity of Greenland blocking (GB) and how the zonal movement of GB influences cold anomalies over North America and Europe. In the introduction, the authors completely ignored the previous studies so that the authors incorrectly said "**To our knowledge, no studies have considered the effect that the propagation velocity of atmospheric blockings has on our weather**". Please the authors read these previous papers.

(3) **Lines 55-56:** "Our study has two objectives: to start with, we will assess the characteristics of the zonal propagation velocity of atmospheric blocks and how it relates to blocking size, intensity, duration". In previous theoretical studies, the zonal propagation velocity of atmospheric blocking linked to the blocking size, intensity and duration has been established in Luo et al. (2019) and Zhang and Luo (2020). They found that a small meridional potential vorticity gradient favors the persistence of atmospheric blocking. When atmospheric blocking is stronger, it shows less eastward movement, larger zonal scale or blocking size and slower decay (Zhang and Luo 2020). What is the difference between the author's results and previous results? Please read the previous papers.

(4)  In the Data and Method section (lines 130-144), the authors tried to use the two-dimensional (2D) Cell-tracking Algorithm to calculate the zonal propagation velocity of atmospheric blocks. However, I think that such a 2D Cell-tracking Algorithm fails to identify the zonal propagation velocity of atmospheric blocking because this 2D Cell-tracking algorithm cannot differentiate the group velocity and phase speed or zonal propagation velocity of the blocking anomaly in the form of

$$\psi_B = B\sqrt{\frac{2}{L_y}}\exp[i(kx-\omega t)]\sin(my)+cc$$ (Luo et al. 2019), where cc denotes the complex conjugate of its preceding term, $L_y$ is the width of beta channel and B(x,t) is the complex blocking envelope amplitude and the time-longitude variation of absolute B(x,t) or $|B|$ denotes the group velocity of the blocking anomaly $\psi_B$ with zonal wavenumber k, $C_p = \omega/k$ is the zonal propagation velocity of the blocking anomaly in a linear theory framework. In a nonlinear theory framework, the zonal propagation velocity of the blocking anomaly is $C_{NP} = U - \dfrac{PV_y}{k^2+m^2+F} - \dfrac{\delta_N M_0^2}{2k PV_y}$ (Luo et al. 2019, JAS), where U is the basic zonal wind, $PV_y$ is the meridional gradient of background potential vorticity and $M_0$ is the blocking amplitude or intensity. If the authors calculate the zonal movement speed of $\psi_B$ by tracking the maximum or minimum intensity of $\psi_B$, this movement speed cannot represent the zonal propagation velocity of atmospheric blocking. Thus, I do not think that the results based on the 2D Cell-tracking algorithm are correct. Please see Zimin et al. (2003, 2006) about how to calculate the group velocity and zonal propagation velocity of atmospheric blocking.

(5) The unit about the zonal propagation velocity of atmospheric blocking. The zonal velocity of Rossby waves is expressed in the unit of "m/s". Thus, I suggest that in Tables 1-2 and Figs. 4, 9, the unit "km/day" should be changed into the unit:"m/s".

(6) There are different zonal movement speeds of atmospheric blockings in different region. The authors should calculate the zonal propagation velocity of atmospheric blocking by dividing the Northern Hemisphere into three (five) regions in winter (summer) according to Fig.2. Unfortunately, the authors did not discuss this issue.

(7) **Line 183**: The authors should clearly describe what do the 10th, 50th, and 90th percentiles mean.

(8) Please explain why the large blocking size tends to be westward-moving and why long-lived or large amplitude blocking tends to be eastward-moving in Fig. 4.

(9) I do not think that the results in Fig. 6 are correct. I do not understand why summer eastward-moving atmospheric blocking events are more frequent in high-latitudes.

In contrast, winter eastward-moving atmospheric blocking events are more frequent in the relatively low latitudes.

(10) In the conclusion section, the authors should also strengthen some comparisons with the previous similar studies.

**References:**

Chen, X., and D. Luo, 2017: Arctic sea ice decline and continental cold anomalies: Upstream and downstream effects of Greenland blocking. Geophys. Res. Lett., 44, doi:10.1002/2016/ GL072387.

Chen, X., et al., 2018: Impact of winter Ural blocking on Arctic sea ice: Short-time variability. J. Climate, 31, 2267-2282.

Yao Y., et al., 2017: Increased quasi-stationarity and persistence of Ural blocking and Eurasian extreme cold events in response to Arctic warming. Part I: Insight from Observational Analyses. J. Climate, 30, 3549-3568.

Luo, D., et al., 2019: A nonlinear theory of atmospheric blocking: A potential vorticity gradient view. J. Atmos. Sci., 76, 2399-2427.

Zhang, W. and D. Luo, 2020: A nonlinear theory of atmospheric blocking: An application to Greenland blocking changes linked to winter Arctic sea ice loss. J. Atmos. Sci., 77, 723-751.

Zimin, A. V., I. Szunyogh, D. J. Patil, B. R. Hunt, and E. Ott, 2003: Extracting envelopes of Rossby wave packets. Mon. Wea. Rev., 131, 1011–1017.

Zimin, A. V., I. Szunyogh, B. R. Hunt, and E. Ott, 2006: Extracting envelopes of nonzonally propagating Rossby wave packets. Mon. Wea. Rev., 134, 1329–1333.

---

## Author Comment (AC1)

**To both referees:**

Thank you to both referees for their constructive comments on the manuscript. A shared critique was the absence of literature regarding the topic of moving atmospheric blocks. The papers mentioned by both referees are of great help and will be used to strengthen the background of the present study and to place it in a better scientific context. Below, answers to specific comments made by the referees can be found.

**Response Anonymous Referee #1:**
*1. Misunderstanding of the quasi-stationarity of blocking.*

We will include more information on how we interpret quasi-stationarity. The three classifications named in the referee's comment are already included in our introduction (see l. 37), but we can expand on this more. In a lot of papers where the blocking index of Tibaldi and Molteni is used, a Eulerian framework is applied, which does not allow analysis of any dynamics in the blocking system. This is the view we wanted to challenge in this study, combined with an impact driven view for which it is more intuitive to look relative to the impact location.

*2. The description on "To our knowledge, no studies have considered the effect that the propagation velocity of atmospheric blockings has on our weather" is not correct. [...] Please the authors read these previous papers*

Thank you for providing these papers. We did not find these when we were looking for papers on the movement of atmospheric blocks. We will include them to improve the background and scientific context of the paper.

*3. What is the difference between the author's results and previous results?*

The most robust feature we see is that stronger westward retrogression is more often seen with blocks with a large blocking size. This is not in contradiction with Luo et al. who state: "When atmospheric blocking is stronger, it shows less eastward movement, larger zonal scale or blocking size and slower decay (Zhang and Luo 2020)". It is reassuring that these results are also found for the GCM ensemble studied here. Furthermore it is very interesting to read that the PV framework is able to explain the results, in terms of underlying properties, e.g. that a weak meridional PV gradient may support long-living 'blocking' features.

*4. [...] Thus, I do not think that the results based on the 2D Cell-tracking algorithm are correct.*

In our approach a blocking 'object' is defined as an atmospheric feature meeting certain characteristics. These definitions are widely applied in literature to diagnose blocking. As such our blocking objects have a zonal velocity, defined as the propagation speed of the center of mass. This pragmatic approach is indeed quite different from a more fundamental wave-theoretical framework in which blocking is defined in terms of interaction of underlying atmospheric waves. In the revised version we will point out these different viewpoints such that no confusion can remain.

*5. The unit about the zonal propagation velocity of atmospheric blocking. The zonal velocity of Rossby waves is expressed in the unit of "m/s". Thus, I suggest that in Tables 1-2 and Figs. 4, 9, the unit "km/day" should be changed into the unit:"m/s".*

The referee suggests to adjust the units of km/day to m/s to align with previous studies. We originally chose to use the unit of km/day, as it gives a better indication of the total distance covered by the block during its lifetime. We do agree that it may be confusing with the more generally used m/s. We will decide on whether we will use both units together or solely use m/s from now on.

*6. There are different zonal movement speeds of atmospheric blockings in different region. The authors should calculate the zonal propagation velocity of atmospheric blocking by dividing the Northern Hemisphere into three (five) regions in winter (summer) according to Fig.2. Unfortunately, the authors did not discuss this issue.*

We discussed the differences in zonal movements speeds in different regions in Figure 6, where we show the spatial distribution of blocks depending on their propagation speed. In this study, we aim to keep a global view on atmospheric blocks, while still showing regional differences by using figures like 2 and 6. We suggest to remake figure 4 based on different zonal regions on the northern hemisphere and comment on any differences we find in the supplements.

*7. Line 183: The authors should clearly describe what do the 10th, 50th, and 90th percentiles mean.*

The percentiles are calculated per blocking characteristic. They show the $90^{th}$, $50^{th}$, and $10^{th}$ percentile of e.g. size per velocity interval, and similar so for the duration and intensity. We will add some information to the text to make this clearer.

*8. Please explain why the large blocking size tends to be westward-moving and why long-lived or large amplitude blocking tends to be eastward-moving in Fig. 4.*

The westward movement is in line with Rossby wave propagation theory. The longer-lived and more intense blocks fall in the quasi-stationary range, with zonal velocities tending toward the east. We do not have an explanation for the last group of eastward-moving large amplitude blocks, as we are still trying to understand this phenomenon.

*9. I do not think that the results in Fig. 6 are correct. I do not understand why summer eastward-moving atmospheric blocking events are more frequent in high-latitudes. In contrast, winter eastward-moving atmospheric blocking events are more frequent in the relatively low latitudes.*

To start, the frequencies are normalised, so direct comparison in their absolute numbers is not possible using this figure. Our hypothesis for the difference between summer and winter is the absence of a mean jet during the summer, while this is present during winter, combined with the shifting minimum latitude of the high-pressure belt with the seasons. We can add a figure to the supplements showing the mean jet using Z500 mean geostrophic velocities. We can also add the total number of blocking events in the upper right corners of each subfigure of Figure 6.

*10. In the conclusion section, the authors should also strengthen some comparisons with the previous similar studies.*

We will add the studies provided by the referee for comparison in the conclusions.

**Response Anonymous Referee #2:**

Introduction: *The authors derive average velocities of the blocks of about 3.5 m/s (300 km/day) which is still much smaller than the typical synoptic scale wind speed of U=10 m/s. From the introduction, the reader gets the impression that the velocities of the block are much higher so that the quasistationarity assumption does not hold. Please clarify.*

In Table 1 we find that absolute zonal mean velocity is approximately 300 km/day or 3.5 m/s. The majority of the blocks have velocities around this value and are categorized as quasi-stationary. However, the 10% fastest moving blocks in both directions exceed the 10 m/s (>864 km/day) and therefore are classified as eastward- or westward moving blocks. We will add the absolute velocities where we now use percentiles to make this clearer, and check the introduction to see where these

assumptions are not stated well enough. We think the title might contribute to the confusion, as it is maybe too generally stated.

Section 2.1 and 2.2.: *Out of curiosity from my side: What method do you use to regrid the data? Is smoothing applied, too? What is your motivation to use these two datasets?*

The motivation for the ECE3p5 and ERA5 datasets is that ERA5 is broadly accepted as the most suitable reanalysis dataset for the midlatitudes on the northern hemisphere, where its biases are limited. The ECE3p5 is the in-house variant of EC-Earth3 and corrected for its biases in the northern hemisphere. We assessed the ability of ECE3p5 to accurately represent atmospheric blockings. The data was regridded using bilinear interpolation with additional smoothing. We will add this to the methods.

Section 2.3: *Your method has a lot of thresholds at several places. How much are the results depending on these settings? Please check that you explain all variables. In the equations you sometimes use t as a coordinate and in other equations d, I assume that it is time in both cases. Why do you change the nomenclature?*

Using "t" and "d" was a typo, as both should be "t" for time. We tested the dependence on the minimum latitude. Results for this are shown in the Supplements. All other threshold follow from the studies cited in this section.

l. 113/eq. (7): *Please explain in more detail the meaning of BI? How can wie interpret the values of BI.*

*l.117: Relating to my former comment: What is the meaning of the different thresholds? Can you give idealized examples?*

The phi underneath eq. 7 is a typo and should not be here. It has not been taken into account in our calculations. The definition of the blocking intensity follows from the paper of Wiedenmann et al. (2002). It is a diagnostic for the relative strength of large-scale flow regimes within blocking regions. The thresholds we used for weak and strong blocking events also come from the paper of Wiedenmann et al. (2002), who based the categorisation on which blocks were within and outside of one standard deviation of the 30-year mean intensity of their dataset. We will add some examples of blocks falling within the different categories.

l. 119/fig. 2: *You speak of blocking intensites, but the figure shows frequencies. Please explain.*

This is a mistake, thank you for noticing. We made two versions of this figure: one with the intensities, and one with the frequency of blocked days, which were exchanged. It should be "Mean BI [-]" and we will change this for the next version.

Section 2.5: *How would your results change if you used maximum or minimum temperature instead of mean temperature?*

We expect using minimum and maximum temperatures would have the most impact in summer for the maximum temperature, as temperature differences between day and night are larger in summer compared to winter. In winter, the minimum temperatures are probably lower. We think this will result in similar patterns, but likely with larger temperature anomalies. We did prepare the minimum and maximum temperatures and could make additional figures for the supplements if needed.

Section 3.1: *How do you define the size of the blocking high? By summing up the grid points? How are winter and summer defined?*

The size of the blocks is defined by the sum of the gridpoints, converted to km$^2$. This is explained in l. 135-138 and eq. 8. Winter and summer are defined as December-January-February, and June-July-August respectively. This is explained in l.142-144.

Table 1: *I would recommend to additionally give the velocities in m/s since it then could be more easily compared to the standard assumption of U=10m/s on the synoptic scale.*

See answer on AR1, Q5.

Section 3.2: *How do you define westward/eastward propagating and quasistationary systems. Please give a clear definition. I am very confused how many systems are in which category and if you define these by certain thresholds or by percentiles. If you use percentiles, please also add the according velocities per category.*

We defined the westward, eastward, and quasi-stationary blockings based on percentiles. We defined $v_x \leq P10$ to be westward moving and $v_x \geq P90$ to be eastward moving. Everything in between $P10 < v_x < P90$ we defined as quasi-stationary. This division was made separately per season. This is explained in section 3.4, but we will check if this needs to be replaced to earlier in the text. We will add the absolute velocities with the percentiles.

Fig. 4: *Can you please add a contour to the frequency shading that represents the percentage of systems: for example does the second blue shading already contain 50% of all systems? If this is not possible at all, please add the probability density distribution or a histogramm of the velocities. I still wonder how many systems are slowly moving and how many systems are faster moving.*

Thank you for this suggestion. We will try to make this clearer in the figure.

Fig. 5: *Some of the data is cut of in the left figure. The 15 day rolling mean is impossible to see, please use a different color, e.g. black, for these lines.*

We will adjust the figure for the cut-off data and change the colour of the 15-day rolling mean to black for better visibility.

l. 235: *Are these the values of the 15 day rolling mean?*

These are the mean values over all blocks and ensembles per day, and not their 15 day rolling mean. We will clarify this in the text.

Fig. 8: *the 40 degree times 80 degree is latitude-longitude, correct? Can you please add composite geopotential height lines here. This could also give you some information on blocking type.*

Yes, the 40x80 is the latitude x longitude. This is a good idea to add the geopotential lines. We will try to add them to the figure.

Table 2: *For the velocities there seem to be very long tails. I would recommend to either add more percentiles (5%,10%, 90%,95%) or add a figure showing the probability density function. See also my comment regarding Fig. 4.*

It is true that the tails are long. If we would add more percentiles to the table, we would also have to add more percentiles to Figure 8 to match. In this figure, you can see that the differences in effect on the temperature is not as large as the jump in velocities. Therefore, we do not think that adding more percentiles to the figure would be beneficial. To option of adding a histogram to Figure 4 is probably more intuitive.

l. 351: *over 2x*

We will remove the double "over".

---

## Referee Report (RR1)

Review result of "On the movement of atmospheric blocking systems and the associated temperature responses" by Mourik et al."

Overall recommendation: Major revision

I have read this revised manuscript in detail. It seems that the authors only made a minor revision and didn't answer my major issues. Thus, I am not satisfied for the author's revision, even though this manuscript has been improved in part. In my initial review, I have pointed out that the author's 2D Cell-tracking Algorithm used to identify the zonal propagation velocity of atmospheric blocking is likely incorrect. Unfortunately, the authors did not present any responses to my issue in this new revised version. Because the 2D Cell-tracking Algorithm failed to identify the zonal propagation speed of atmospheric blocking, the author's conclusion about the propagation speed of atmospheric blocking is confusing and not solid. Based on this, I recommend that a major revision of this manuscript is still needed.

Major comments:

(1) In my initial review, I have pointed out that the two-dimensional (2D) Cell-tracking Algorithm as a Lagrangian method cannot correctly identify the zonal propagation speed of atmospheric blocking. Such a method will lead to a misleading conclusion. Unfortunately, the authors did not present any responses in the revised manuscript. Below, I again give my issues:

In the Data and Method section (lines 130-144 in the original manuscript), the authors tried to use the two-dimensional (2D) Cell-tracking Algorithm to calculate the zonal propagation velocity of atmospheric blocks. However, I think that such a 2D Cell-tracking Algorithm fails to identify the zonal propagation velocity of atmospheric blocking because this 2D Cell-tracking algorithm cannot differentiate the group velocity and zonal phase speed or propagation velocity of the blocking anomaly in the geopotential height fields in the form of

$$\psi_B = B\sqrt{\frac{2}{L_y}}\exp[i(kx - \omega t)]\sin(my) + cc$$ (Luo et al. 2019), where cc denotes the

complex conjugate of its preceding term, $L_y$ is the width of beta channel and B(x,t) is

the complex blocking envelope amplitude and the time-longitude variation of absolute B(x,t) or $|B|$ denotes the group velocity of the blocking anomaly $\psi_B$ with zonal wavenumber k, $C_p = \omega/k$ is the zonal propagation velocity of the linear blocking anomaly in a linear theory framework. In a nonlinear theory framework, the zonal propagation velocity of the blocking anomaly is $C_{NP} = U - \dfrac{PV_y}{k^2 + m^2 + F} - \dfrac{\delta_N M_0^2}{2k PV_y}$,

where U is the basic zonal wind, $PV_y$ is the meridional gradient of background potential vorticity and $M_0$ is the blocking amplitude or intensity. If the authors calculate the movement speed of $\psi_B$ by tracking the maximum or minimum intensity of $\psi_B$, this movement speed cannot represent the zonal propagation velocity of atmospheric blocking. Thus, I do not think that the results based on the 2D Cell-tracking algorithm are correct. Of course, it is also difficult to compute the group velocity and zonal propagation velocity of atmospheric blocking using the 2D Cell-tracking algorithm. Please also see Zimin et al. (2003, 2006) about how to calculate the group velocity and zonal propagation velocity of Rossby wave packet (atmospheric blocking).

Based on the blocking theory, I think that the 2D Cell-tracking algorithm the authors used are not correct. The authors should at least provide evidence to confirm if the author's method is correct. For example, in a revised version the authors should at least discuss some issues: For example, the authors should discuss whether the 2D Cell-tracking algorithm is suitable for identifying the zonal propagation speed of atmospheric blocking。Is whether it consistent with the previous blocking theory (i.e., Luo et al. 2019) and the previous diagnostic methods (i.e., Zimin et al. 2003, 2006). I suggest that the authors should calculate the zonal movement speeds of quasi-stationary, retrograde and eastward-moving blocking events by plotting the time-longitude Hovmoller diagrams of the composite daily Z500 anomalies for each type of blocking events and then further compare them with the results obtained from 2D Cell-tracking algorithm. Such a comparison can confirm whether the 2D Cell-tracking algorithm is correct.

(2) How to calculate the zonal phase speed of blocking events

When blocking events in a certain region are classified into three types: Quasi-stationary, retrograde and eastward-travelling blocking events, the authors may perform a daily composite to the daily evolution of the Z500 (t, x, y) anomaly for each type of blocking events. The authors can calculate the zonal phase speed of blocking for each type of blocking event if the authors plot the time-longitude Hovmoller diagram of the composite daily Z500 (t, x, y) at a fixed latitude y. However, when the authors used the 2D Cell-tracking algorithm to calculate the movement speed by tracking the maximum or minimum value of daily Z500 (t, x,y), the movement speed obtained by the authors is not identical to the zonal phase speed of blocking.

(3) Whether atmospheric blocking undergoes retrogression, westward or eastward movement depends on the background zonal wind U, $PV_y$ and blocking amplitude or intensity in terms of $C_{NP} = U - \dfrac{PV_y}{k^2 + m^2 + F} - \dfrac{\delta_N M_0^2}{2k PV_y}$ (Luo et al. 2019). When blocking intensity or amplitude is larger or U or PVy is smaller, atmospheric blocking exhibits a westward movement. In particular, atmospheric blocking with a large zonal size often shows a retrogression because it has generally a large amplitude or intensity.

However, the 2D Cell-tracking algorithm seems to give confused results. For example, in Table 1, it seems that blocking events with large size and intensity move more eastward with a large phase speed. This is very confusing. I suggest that the authors should divide blocking events in winter or summer into three cases: Stationary, retrogression and eastward movement. Then, the authors should calculate the event number, size, intensity and phase speed of the three types of blocking events in winter or summer and further discuss their impacts on local weathers. The authors may find that retrograde blocking events will have larger size (zonal scale), larger intensity, and longer lifetime, which can be more stationary and eastward-travelling if the upstream basic zonal wind is particularly strong (Zhang and Luo 2020). In contrast, the eastward-traveling blocking events will generally have lower intensity, smaller size and shorter lifetime. The authors should read in

detail the theoretical papers of Luo et al. (2019) and Zhang and Luo (2020). They have discussed in detail the relationship among the blocking size, intensity and movement as well as their linkages to the background conditions and their impacts.

(4) Please clearly give the definition of the blocking duration. In fact, the duration of atmospheric blocking has different definitions. For example, the persistent time of blocking anomaly with amplitude larger than a threshold in a given domain may be defined as a local duration. This local duration is not equivalent to the lifetime of blocking when the blocking moves in the west-east direction. Clearly, the local duration and lifetime of blocking events are different. It seems that the local duration of the blocking events is more important for the local weather extremes than its lifetime unless the blocking is stationary.

Minor issues are not further shown here.

References:

Luo, D., et al., 2019: A nonlinear theory of atmospheric blocking: A potential vorticity gradient view. J. Atmos. Sci., 76, 2399-2427.

Zimin, A. V., I. Szunyogh, D. J. Patil, B. R. Hunt, and E. Ott, 2003: Extracting envelopes of Rossby wave packets. Mon. Wea. Rev., 131, 1011–1017.

Zimin, A. V., I. Szunyogh, B. R. Hunt, and E. Ott, 2006: Extracting envelopes of nonzonally propagating Rossby wave packets. Mon. Wea. Rev., 134, 1329–1333.

---

## Author Response (AR2)

- Original comments in black
- Answers in blue

We thank the reviewers and editor for their time and constructive feedback to improve the manuscript.

**Report #1:**

The paper investigates the motion of atmospheric blocking and its influence on the temperature field. The paper has improved from the first round of reviews; I especially like the discussion section that explores possible influences of the method on the data.

I still have a few points:

- Please indicate in each figure to which data the figure belongs (ERA5 or ECE3p5). In the later figures, this is not clearly labeled. If it is based on ECE3p5 and you have similar analysis for ERA5, please plot them or their differences in the supplement.

Answer: We added the datasets where missing in the figures. From figure 4 onwards, all figures are based on ECE3p5. ERA5 is only used to verify the ability of ECE3p5 to simulate blocking events, but due to the lower amount of data (1950-2022 compared to 16 times 1850-2014 in ECE3p5) we only made the composite plots with the ECE3p5 data.

- Regarding Fig. 2: Can you speculate why the frequency of blocks in the ECE3p5 data is shifted compared to ERA5?

Answer: Yes, this shift of frequencies is explained in the discussion 4.1 "Model biases" in the last paragraph (lines 398-405). We refer to this explanation in the last sentence of Section 2.3. In summary, the shift is likely explained by a combination of overestimation of jets over the oceans and inaccurate orography in climate models.

- Section 2.4: You speak of intensity and size. Please clarify if you mean mean or maximum BI as intensity. What is meant by size: the maximum size during the lifetime or on a certain day of the block? Moreover, please clarify that the size is dependent on your method of blocking identification. The whole block, as well as its area of influence, can be larger. Your method basically identifies the blocking high central area only.

Answer: Thank you for pointing out this confusion in definitions. Both the mean and maximum intensity are calculated per block per day. For every figure or description it is stated which of the two is used (however, maximum intensity is only used in Table 1). The size is also calculated per day of the blocking event. We will clarify this in the Methods (section 2.3 and 2.4) and elsewhere needed. It indeed finds the blocking high central area, as can be seen in Figure 3, and follows from equations 1-6, which is the same as Tibaldi et al. and Sousa et al. used, so this area is what has been referred to as a block in these papers as well.

- Section 2.4: Elaborate on how you calculate the weighted center (based on BI values?).

Answer: the weighted center is based on the blocking intensity and is a direct output of the 2D-celltracking algorithm. Every cell that is part of the block has an intensity, and from all these intensities the weighted center is calculated, resulting in a (lat,lon) output per day that the block exists.

- Section 2.4: To calculate the speed of the block centers, you use the location at the first day of identification and the last day. I wonder if this would be different if you used a few time steps in between. It might be possible that the systems move faster at the beginning and at the end of their "lifetime."

Answer: Yes, we also find that blocks move faster in their first days. This is why Figure 6 is plotted for the fourth day, which is also explained in lines 273-277 in Section 3.4 and shown by Steinfeld et al. (2018). However, when you look at Figure S5 (newly added in the supplements), we can see that the movement continues after the initial onset phase. We will also refer to this new figure in the text and add the logic behind it in the methods.

- Figure 3: Could you please add a date to the figure? Is this data from ERA5?

Answer: Date is added in the figure. It is indeed from ERA5. We also added a figure to better show how the algorithm captures the development of the block. The arrow was accidently pointed in the wrong direction (based on assumptions on the general blocking direction), so this has also been adjusted for this specific case.

- Section 2.5, line 148: How is the data exactly "detrended and corrected for climatology"?

Answer: This is done by calculating the climatological daily mean of the temperature per ensemble member and subtracting this mean from each day in the dataset. We will add this to the text.

- Section 3.1, lines 180–181: Hirt et al. (2018, their Fig. 7, Section 4.1.5) investigated blocking from a point vortex perspective and found that most of the identified, theoretical, westward-directed blocking speeds are smaller in magnitude than the westerly flow. Can you confirm this? How many of your blocks are moving to the east? [Reference: Hirt, M., Schielicke, L., Müller, A., & Névir, P. (2018). Statistics and dynamics of blockings with a point vortex model. *Tellus A: Dynamic Meteorology and Oceanography, 70*(1), 1–20. https://doi.org/10.1080/16000870.2018.1458565]

Answer: This is interesting and I will add the reference to this section of the results. However, we didn't study the wind velocities in this work, so we can't say anything about their relative magnitudes. As for how many of our blocks are moving to the east and west, we split it out here in two tables for ERA5 and EC-Earth3p5 separately. However, as this just splits the velocity in positive and negative velocities, of which the majority is still around zero (see Figures 4, 7 and Table 2, where the distributions of blocking velocities are shown) we think it is more useful to work with the fastest 10% moving eastward and westward (shaded in grey in those figures). These are also the percentiles by which the blocks are divided in Figure 6.

| ERA5 | Total (1951-2014) | $v_x>0$ | $v_x<0$ | $v_x=0$ |
|---|---|---|---|---|
| Full year | 7511 | 5102 (68%) | 2401 (42%) | 8 |
| DJF | 1557 | 981 (63%) | 576 (37%) | 0 |
| JJA | 2273 | 1514 (67%) | 754 (33%) | 5 |

| EC-Earth3p5 | Total (1951-2014, 16 ens. members) | $v_x>0$ | $v_x<0$ | $v_x=0$ |
|---|---|---|---|---|
| Full year | 109994 | 70144 (64%) | 39820 (36%) | 30 |
| DJF | 25139 | 15680 (62%) | 9451 (38%) | 8 |
| JJA | 28816 | 17806 (62%) | 11001 (38%) | 9 |

- Fig. 4: Size and intensity—are these mean values over the lifetime of each block?

Answer: Good question. Both are the mean values over the lifetime of a block. We will clarify this in the text.

- Fig. 5: Black lines are not explained in the caption.

Answer: The lines are not black, but a darker version of each colour. In the description, we described it as "The **thicker** lines are the 15-day rolling means, and the **thinner** lines show the mean values over all blocks and ensembles per day of the year." We changed it to "darker" and "lighter", which is hopefully clearer.

- Section 3.3, line 225: A definition of westward-/eastward-moving blocks is missing. The definition follows in Section 3.4.

Answer: We adjusted the text in several places revering to westward- and eastward-moving blocks to distinguish between all blocks moving to the east/west or the 10% fastest moving blocks. In 3.3 the 10% fastest is not used and the blocks are merely split by their positive or negative velocity.

- Fig. 6: The hotspots of the distributions are mainly far to the north. Can you explain this? How would the distributions be affected if you take all grid points into consideration, not only the centers? Could you please add the latitude you use as a criterion for the identification?

Answer: This is a good question, especially for summer when the blocks are found even further to the north. The latitudes we use as criteria can be found in the methods (Section 2.3, equations 1 and 3. Minimum latitude is variable, but around 40 degrees, and the maximum latitude is 75 degrees). We will adjust the methods to make it clearer which maximum latitude we use there. It follows from Sousa et al. (2021). Just as there is a high-pressure band just below the minimum latitude, there is also a high pressure area over the poles, which is why the maximum latitude is applied to exclude those. In Sousa et al. they call blocks forming in this area "polar blocks", and in their figure 7 it is visible that Rex blocks occur in these regions quite often. It is thus possible that what we see around the maximum latitude are part of a bigger blocking system over the pole. We will expand on this in the manuscript.

- Fig. 8: On which day do you calculate the composites? Are the descriptions of the x- and y-axis correct in the caption?

Answer: The composites are calculated over the whole duration of the block, we will add this in the description. Thank you for pointing out the error in the caption. Originally the x- and y-axes where switched, which caused the error. We changed it to the correct description.

- Section 5, lines 490–493: I was also wondering if we can still call these fast-moving systems blocking. Could you look at some examples explicitly and make a statement on this?

Answer: This question does not have a straight-forward answer. Based on the widely used blocking definition using Z500, these fast-moving systems classify as blocks, and according to figure 8 these systems also still impact the temperature at the Earth's surface. Therefore you could say that at least their moving impact is still block-like. However, when looking from a stationary point of view (e.g. one location on Earth), these systems will likely result in a lower impact due to the faster passing time and may therefore not be experienced as a classical block.  This is mentioned in the last paragraph of 4.3 and 5, lines 513-515, but we will add a concluding section in the last paragraph of 5.

**Report #2:**

Review result of "On the movement of atmospheric blocking systems and the associated temperature responses" by Van Mourik et al."

Overall recommendation: Major revision

I have read this revised manuscript in detail. It seems that the authors only made a minor revision and didn't answer my major issues. Thus, I am not satisfied for the author's revision, even though this manuscript has been improved in part. In my initial review, I have pointed out that the author's 2D Cell-tracking Algorithm used to identify the zonal propagation velocity of atmospheric blocking is likely incorrect. Unfortunately, the authors did not present any responses to my issue in this new revised version. Because the 2D Cell-tracking Algorithm failed to identify the zonal propagation speed of atmospheric blocking, the author's conclusion about the propagation speed of atmospheric blocking is confusing and not solid. Based on this, I recommend that a major revision of this manuscript is still needed.

Answer: We thank the reviewer again for their detailed comments and suggestions. The major concerns with the methodology probably lie in our motivation and terminology, so we suggest to re-phrase several aspects.

Firstly, we want to stress that the focus of this work is not to compare the movement of atmospheric blocks to the theoretical propagation speed of Rossby waves. Although we agree that a more detailed and quantitative comparison to Rossby wave theory should help with the interpretation of the results, this is not the main goal of our work. Rather, we want to study the actual movement with respect to the Earth's surface linked to the potential impacts on near-surface conditions. While there is undoubtedly value in linking surface-relative movement to the group speed of associated Rossby waves, it is not of primary importance to our results. We still agree that this should be communicated and motivated more clearly and therefore suggest to make the necessary adjustments to the text as well as use the terminology (surface-relative) 'movement' rather than zonal 'propagation speed'.

Secondly, the reviewer does not agree with the 2D cell-tracking algorithm used to detect and track atmospheric blocks. While the technique certainly has its limitations (see Discussion section 4.2) and the link to a more formal interpretation using Rossby wave theory is not straightforward, we refer the reader to the work of Lochbihler et al. (2017) and others on this matter, as it is an existing method to track movements in for example rainfall events and other moving "objects" in the atmosphere. In addition, our suggestion is to improve the link to Rossby wave theory in order to better estimate wave propagation speeds and compare these to the tracked movements of blocks. This is done in the results section 3.2.

Lastly, an argument not to rely solely on Rossby wave theory is that the properties of atmospheric blocks considered here cannot be fully represented as such. Even including nonlinear terms, complex wave-wave interactions and highly nonlinear behaviour may be of importance while they are not captured in an idealised theoretical view. Again, we suggest to connect better to existing theoretical understanding but stick to the observational, surface-relative view of atmospheric blocks as our primary viewpoint.

Major comments:

(1) In my initial review, I have pointed out that the two-dimensional (2D) Cell-tracking Algorithm as a Lagrangian method cannot correctly identify the zonal propagation speed of atmospheric blocking. Such a method will lead to a misleading conclusion. Unfortunately, the authors did not present any responses in the revised manuscript. Below, I again give my issues:

In the Data and Method section (lines 130-144 in the original manuscript), the authors tried to use the two-dimensional (2D) Cell-tracking Algorithm to calculate the zonal propagation velocity of atmospheric blocks. However, I think that such a 2D Celltracking Algorithm fails to identify the zonal propagation velocity of atmospheric blocking because this 2D Cell-tracking algorithm cannot differentiate the group velocity and zonal phase speed or propagation velocity of the blocking anomaly in the geopotential height fields in the form of

$$\psi_B = B\sqrt{\frac{2}{L_y}}\, exp[i(kx - \omega t)]sin(my) + cc$$

(Luo et al. 2019), where cc denotes the complex conjugate of its preceding term, Ly is the width of beta channel and B(x,t) is the complex blocking envelope amplitude and the time-longitude variation of absolute B(x,t) or |B| denotes the group velocity of the blocking anomaly $\psi B$ with zonal wavenumber k, Cp=$\omega$/k is the zonal propagation velocity of the linear blocking anomaly in a linear theory framework. In a nonlinear theory framework, the zonal propagation velocity of the blocking anomaly is $C_{NP} = U - \frac{PV_y}{k^2+m^2+F} - \frac{\delta_N M_0^2}{2kPV_y}$

where U is the basic zonal wind, PVy is the meridional gradient of background potential vorticity and M0 is the blocking amplitude or intensity.

If the authors calculate the movement speed of $\psi B$ by tracking the maximum or minimum intensity of $\psi B$, this movement speed cannot represent the zonal propagation velocity of atmospheric blocking. Thus, I do not think that the results based on the 2D Cell-tracking algorithm are correct. Of course, it is also difficult to compute the group velocity and zonal propagation velocity of atmospheric blocking using the 2D Cell-tracking algorithm. Please also see Zimin et al. (2003, 2006) about how to calculate the group velocity and zonal propagation velocity of Rossby wave packet (atmospheric blocking).

Based on the blocking theory, I think that the 2D Cell-tracking algorithm the authors used are not correct. The authors should at least provide evidence to confirm if the author's method is correct. For example, in a revised version the authors should at least discuss some issues: For example, the authors should discuss whether the 2D Celltracking algorithm is suitable for identifying the zonal propagation speed of atmospheric blocking is whether it consistent with the previous blocking theory (i.e., Luo et al. 2019) and the previous diagnostic methods (i.e., Zimin et al. 2003, 2006). I suggest that the authors should calculate the zonal movement speeds of quasi-stationary, retrograde and eastward-moving blocking events by plotting the time-longitude Hovmoller diagrams of the composite daily Z500 anomalies for each type of blocking events and then further compare them with the results obtained from 2D Cell-tracking algorithm. Such a comparison can confirm whether the 2D Cell-tracking algorithm is correct.

Answer:

- We thank the reviewer for these detailed theoretical considerations and suggestions. We fully agree that there is a difference between the formal definition of group and phase velocity of Rossby wave packets, and the heuristic definition of the 'propagation velocity' we use, namely 'the movement of the intensity-weighted centre of the tracked blocking feature'. Our definition is grounded in a pragmatic approach where a block is first identified by its 'blocking index'. Since this index is based on inherently simple threshold criteria it will and cannot correspond to a definition based on wave theory. Our main motivation for using a simple definition based on the tracked features rather than

on the underlying propagating Rossby wave, is that our interest is in the associated temperature response. Therefore, to us defining a blocking centre as the weighted mean centre of the tracked feature is useful because it allows us to make composites, and similarly a notion of the movement of the whole system relative to the ground is useful as it allows us to investigate the differences in temperature response they are associated with. To avoid further confusion of the term, we removed the "propagation" part from our definition of blocking movement in the text and changed it to "zonal velocity" and "blocking movement".

To incorporate more of the requested information provided by the reviewer, we expand a bit more on the theoretical propagation velocity in the introduction, and on the Rossby Wave theory in the results, quoting the formulae introduced above. In addition we will include some Hovmöller diagrams which show how the movement of the atmospheric blocking patterns is extracted and recorded for one case in Figure 3 and for the composite of all velocities <P10, between P10 and P90, and <P90 in the supplements Figure S5. From these figures it can be seen that the intensity-weighted centres follow the Z500 anomalies. We consider a more detailed comparison and decomposition of each blocking feature in terms of its underlying wave components, beyond the scope of this paper.

(2) How to calculate the zonal phase speed of blocking events

When blocking events in a certain region are classified into three types: Quasistationary, retrograde and eastward-travelling blocking events, the authors may perform a daily composite to the daily evolution of the Z500 (t, x, y) anomaly for each type of blocking events. The authors can calculate the zonal phase speed of blocking for each type of blocking event if the authors plot the time-longitude Hovmoller diagram of the composite daily Z500 (t, x, y) at a fixed latitude y. However, when the authors used the 2D Cell-tracking algorithm to calculate the movement speed by tracking the maximum or minimum value of daily Z500 (t, x,y), the movement speed obtained by the authors is not identical to the zonal phase speed of blocking.

Answer: We adjusted Figure 3 to include a Hovmöller diagram of the Z500 anomaly, the longitudes of the intensity-weighted centres following from the tracking algorithm, and the boundaries of the block following from the blocking index. Additionally, we added a section in the supplements with composite Hovmöller plots (S5) for all velocities lower that P10, between P10-P90, and above P90 (similar to Figure 6)  and extra explanation. These dots in these figures follow from the intensity-weighted mean, which is something different than the minimum or maximum value of daily Z500 as the reviewer is suggesting. The intensity is calculated by comparing Z500(t,x,y) to Z500 upstream and downstream (see equation 7) and is thus a relative value with respect to its surroundings.

(3) Whether atmospheric blocking undergoes retrogression, westward or eastward movement depends on the background zonal wind U, PVy and blocking amplitude or intensity in terms of $C_{NP} = U - \frac{PV_y}{k^2+m^2+F} - \frac{\delta_N M_0^2}{2kPV_y}$ (Luo et al. 2019). When blocking intensity or amplitude is larger or U or PVy is smaller, atmospheric blocking exhibits a westward movement. In particular, atmospheric blocking with a large zonal size often shows a retrogression because it has generally a large amplitude or intensity. However, the 2D Cell-tracking algorithm seems to give confused results. For example, in Table 1, it seems that blocking events with large size and intensity move more eastward with a large phase speed. This is very confusing. I suggest that the authors should divide blocking events in winter or summer into three cases: Stationary, retrogression and eastward movement. Then, the authors should calculate the event number, size, intensity and phase speed of the three types of blocking events in winter or summer and further discuss their impacts on local weathers. The authors may find that retrograde blocking events will have larger size (zonal scale), larger

intensity, and longer lifetime, which can be more stationary and eastward-travelling if the upstream basic zonal wind is particularly strong (Zhang and Luo 2020). In contrast, the eastward-traveling blocking events will generally have lower intensity, smaller size and shorter lifetime. The authors should read in detail the theoretical papers of Luo et al. (2019) and Zhang and Luo (2020). They have discussed in detail the relationship among the blocking size, intensity and movement as well as their linkages to the background conditions and their impacts.

Answer: Thank you for this extra information which we will incorporate to strengthen our explanations. However, Table 1 seems to be misunderstood. It shows the statistics over all blocks, only divided by their seasons, to compare the numbers of the used climate model to the ERA5 reanalysis. The velocity in this table is the absolute velocity and thus does not indicate any west- or eastward movement. The distinction between the different movement directions is made for the first time in Figure 4, where indeed you can see that the westward-moving blocks are larger in size and the eastward-moving blocks smaller. The quasi-stationary blocks are mid-sized, but have the longer lifetime and intensity. This agrees with what you are describing here.

(4) Please clearly give the definition of the blocking duration. In fact, the duration of atmospheric blocking has different definitions. For example, the persistent time of blocking anomaly with amplitude larger than a threshold in a given domain may be defined as a local duration. This local duration is not equivalent to the lifetime of blocking when the blocking moves in the west-east direction. Clearly, the local duration and lifetime of blocking events are different. It seems that the local duration of the blocking events is more important for the local weather extremes than its lifetime unless the blocking is stationary.

Answer: Yes, the local duration and the lifetime of the block are different. In this manuscript we look at the lifetime of the blocks, as we are working in a Lagrangian framework and thus are following the blocks. Since we don't work from a local (Eulerian) point of view, it is not possible to give a general local duration of a block. We will clarify this in the methods and elsewhere where "duration" is used.

Minor issues are not further shown here.

References:

Luo, D., et al., 2019: A nonlinear theory of atmospheric blocking: A potential vorticity

gradient view. J. Atmos. Sci., 76, 2399-2427.

Zimin, A. V., I. Szunyogh, D. J. Patil, B. R. Hunt, and E. Ott, 2003: Extracting envelopes

of Rossby wave packets. Mon. Wea. Rev., 131, 1011–1017.

Zimin, A. V., I. Szunyogh, B. R. Hunt, and E. Ott, 2006: Extracting envelopes of

nonzonally propagating Rossby wave packets. Mon. Wea. Rev., 134, 1329–1333

**Editor:**

Despite some improvements of the paper, the two reviewers and myself have still several major and minor comments. Reviewer 1 would like to be convinced about the relevance of the propagation velocity computation using the cell-tracking algorithm. Reviewer's idea is to compare the propagation velocity done in the paper with that obtained from an Hovmoller diagram on the composite of anomalous Z500 over blocking events. One possibility would be to show several

individual blocking cases rather than a composite. Reviewer 2's comments ask for more detailed information on several figures and results.

On my side, I also noticed that the robust link found in the paper between propagation velocity and size is not fully explained. Line 198 it is said "This result is similar to the behaviour of Rossby waves according to the Rossby wave theory, where the group velocity is westward for larger waves and eastward for smaller waves". First "group velocity" needs to be replaced by "phase velocity". And also I think it would be worth to make the link clearer by explicitly writing the phase velocity in Rossby waves theory (like the one written by reviewer 1). It shows that when the zonal wavenumber is small the PV gradient term, which is negative, is also stronger in amplitude.

Answer: Thank you for your this overview of the reviewers points and your additional comments.

We hope that the addition of extra explanations on our definition of blocking movement and the link to the theoretical propagation and Rossby wave theory improved this paper sufficiently. All the captions of the figures have been checked and adjusted where confusion was still possible. To clarify the tracking algorithm, we added a subfigure to Figure 3, in which a Hovmöller diagram is shown for Z500 anomaly and the boundary of the block over the lifetime of the block used in this figure. Additionally, we added a section in the supplements with Hovmöller plots of the composites of blocks with velocities <P10, between P10 and P90, and >P90. We refer to this figure throughout the text whenever it clarification was needed. We also sized up Figure 8, as we thought it might be a bit small originally.

We will await the further decisions regarding the progression of this paper.

---

## Author Response (AR3)

- Answers in blue
- Original comments in black

Overall recommendation: A minor revision

This manuscript has been greatly improved compared to its first revision. However, I still found that there are some minor issues in this revised version that should be revised.

Thank you for taking the time to review this manuscript again (and so fast!). We took care to incorporate all last points into the final manuscript.

Minor comments:

☒(1) "Summer (1963)" on lines 40-45 of page 2 should be changed into "Summer (1959)"

- Answer: Changed

☒(2)" Luo and Zhang (2020)" on lines 50-55 of page 2 should be changed into "Luo et al. (2019)".

- Answer: Changed

☒(3) "Where Sousa et al. (2021) " on lines 105-110 of page 4 should be changed into "While Sousa et al. (2021)".

- Answer: Changed

☒(4) "this distance" on lines 160-165 of page 6 should be changed into "this zonal distance".

- Answer: Changed

☒(5) "geopotential height" on lines 420-425 of page 20 should be changed into "geopotential height anomalies".

- Answer: Changed

☒(6) "Another noteworthy discovery..." on lines 515-520 of page 24. These results are consistent with the observational and theoretical findings of Zhang and Luo (2020). Please read their paper in detail. Maybe, the authors can use the theoretical results of Zhang and Luo (2020) to make a physical explanation about the author's results. Zhang and Luo (2020) found that a smaller meridional basic potential vorticity gradient (PVy) background favors westward-moving blocking with both larger size and intensity. For this case, the blocking system has weaker dispersion and stronger nonlinearity. Thus, there is a nonlinear relationship among the blocking size, intensity and westward-moving velocity. A large PVy condition favors eastward-moving blocks. For this case, the blocking system has stronger dispersion and weaker nonlinearity. Thus, the relationship among the blocking size, intensity and eastward-moving velocity is approximately linear. Of course, PVy can be influenced by sea-ice extent, SST anomalies and other climatic factors.

- Answer: We added this extra explanation at the end of Section 3.2, where we think it is more fitting.

☒(7) "Zhang and Luo (2019)" on lines 525-530 of page 24 should be changed into "Zhang and Luo (2020)".

- Answer: Changed

☒(8) Line 645, "1963" should be changed into "1959"

- Answer: Changed

☒(9) Line 655, "2019" should be changed into "2020".

- Answer: Changed